# Droplet-based high-throughput single microbe RNA sequencing by smRandom-seq

Ziye Xu[1,2,9], Yuting Wang[2,3,9], Kuanwei Sheng ®[4,5,9] ✉, Raoul Rosenthal[6], Nan Liu ®[2], Xiaoting Hua ®[7], Tianyu Zhang[2], Jiaye Chen ®[8], Mengdi Song[2], Yuexiao Lv[2], Shunji Zhang[3], Yingjuan Huang[2], Zhaolun Wang[2], Ting Cao ®[1,4,6], Yifei Shen[1], Yan Jiang[7], Yunsong Yu ®[7], Yu Chen[1], Guoji Guo ®[2], Peng Yin ®[4,5] ✉, David A. Weitz ®[4,6] ✉ & Yongcheng Wang ®[1,2,3,4] ✉

Bacteria colonize almost all parts of the human body and can differ significantly. However, the population level transcriptomics measurements can only describe the average bacteria population behaviors, ignoring the heterogeneity among bacteria. Here, we report a droplet-based high-throughput single-microbe RNA-seq assay (smRandom-seq), using random primers for in situ cDNA generation, droplets for single-microbe barcoding, and CRISPR-based rRNA depletion for mRNA enrichment. smRandom-seq showed a high species specificity (99%), a minor doublet rate (1.6%), a reduced rRNA percentage (32%), and a sensitive gene detection (a median of ~1000 genes per single *E. coli*). Furthermore, smRandom-seq successfully captured transcriptome changes of thousands of individual *E. coli* and discovered a few antibiotic resistant subpopulations displaying distinct gene expression patterns of SOS response and metabolic pathways in *E. coli* population upon antibiotic stress. smRandom-seq provides a high-throughput single-microbe transcriptome profiling tool that will facilitate future discoveries in microbial resistance, persistence, microbe-host interaction, and microbiome research.

Only recently have scientists begun to appreciate how much our health and disease depend on the trillions of resident microbial communities[1]. Microbial transcriptomics has been a powerful tool for understanding the complexity, plasticity, and regulatory mechanism of microbes[2,3]. However, these population-level gene expression measurements can only describe the average microbe population behaviors. Individual microbes, even of an isogenic population, can keep evolving and differ significantly to increase the adaptivity to environmental changes[4]. To reveal the microbial states with different gene expression patterns, which is critical for microbial resistance,

persistence, microbe–host interaction, and microbiome research, transcriptome profiling of each microbe is required.

Since its establishment in 2009[5], multiple single-cell RNA-seq (scRNA-seq) methods have been developed and widely used in eukaryotic biology[6–10]. The key strategy of current prevalent droplet-based scRNA-seq systems, such as 10X Genomics Chromium platform[7], is using droplet microfluidics to co-encapsulate a single cell with a uniquely DNA barcoded bead in a droplet, where subsequently polyadenylated mRNAs are captured by billions of copies of unique barcoded poly(T) primers carried on the bead. However, this strategy is

[1]Department of Laboratory Medicine, the First Affiliated Hospital, Zhejiang University School of Medicine, Hangzhou, China. [2]Liangzhu Laboratory, Zhejiang University, Hangzhou, China. [3]College of Biomedical Engineering and Instrument Science, Zhejiang University, Hangzhou, China. [4]Wyss Institute for Biologically Inspired Engineering, Harvard University, Boston, USA. [5]Department of Systems Biology, Harvard Medical School, Boston, MA, USA. [6]John A. Paulson School of Engineering and Applied Sciences and Department of Physics, Harvard University, Cambridge, MA, USA. [7]Department of Infectious Diseases, Sir Run Run Shaw Hospital, Zhejiang University School of Medicine, Hangzhou, China. [8]Department of Biomedical Informatics, Harvard Medical School, Boston, MA, USA. [9]These authors contributed equally: Ziye Xu, Yuting Wang, Kuanwei Sheng. ✉e-mail: Kuanwei.Sheng@wyss.harvard.edu; Peng_Yin@hms.harvard.edu; weitz@seas.harvard.edu; yongcheng@zju.edu.cn

incompatible with the single-cell RNA-seq of bacteria. One major reason is that bacterial mRNAs lack 3'-end poly(A) tails, the capture site for poly(T) primers. In addition, the RNA content of a typical bacterium is about two orders of magnitude lower than a typical mammalian cell[3,11], and ribosomal RNA (rRNA) occupies >80% of total bacterial RNAs[12], which makes it challenging to efficiently capture and differentiate those low content mRNAs in single bacterium level. The tough cell wall also makes it hard to lyse bacteria and release the enclosed RNA in the droplet. Some relatively low-throughput single bacterium RNA-seq methods[13,14] have been developed to address these issues, which require single bacterium isolation into multi-well plates using single cell manipulator[13] or FACS[14]. Two previously developed bacterial scRNA-seq methods, PETRI-seq[15], and microSPLiT[16], using random reverse transcription (RT) primers, enable researchers to analyze thousands of fixed bacteria simultaneously with a split-pool barcoding strategy[17], which have made great advances in our ability to study the transcriptome heterogeneity of bacterial communities on a large scale. Nonetheless, this combinatorial split-pool barcoding strategy requires multiple reactions in 96-well plates, which is not a popular or sensitive platform for high-throughput single-cell RNA-seq. Moreover, the majority of mapped reads in both PETRI-seq[15] and microSPLiT[16] were rRNA even with mRNA enrichment, leading to considerable sequencing costs.

Microfluidic has been widely used for high-throughput single-cell analysis[18,19]. Our previous work has established a microfluidic droplet barcoding platform and developed inDrop[20], a high-throughput method for single-cell RNA sequencing, and Microbe-seq[21], a high-throughput method for single-microbe genome sequencing.

In this work, we advance on recent efforts to develop a high-throughput and high-sensitive single microbe RNA-seq method (smRandom-seq), using random primers for in situ complementary DNA (cDNA) generation, droplets microfluidics for single-microbe barcoding and CRISPR-based rRNA depletion for mRNA enrichment (Fig. 1). The droplet microfluidics platform shows high barcoding efficiency, high species specificity (99%), minor cross-contamination (1.6%), and automation capability. The CRISPR-based rRNA depletion dramatically reduces the rRNA percentage (83–32%). The mapped mRNA reads of *E. coli* are highly enriched with a 4-fold increase (16–63%) due to efficient rRNA depletion. A median of ~1000 genes can be detected per bacterium in a population of ~8000 *E. coli*. We also demonstrate that smRandom-seq could be applied to other common bacterial species, including Gram-negative bacteria (*A. baumannii*, *K. pneumoniae*, and *P. aeruginosa*) and Gram-positive bacteria (*B. subtilis* and *S. aureus*). Moreover, we find that *E. coli* population displays morphologic heterogeneity upon antibiotic stress and perform smRandom-seq to investigate these transcriptome changes of the individual bacterium. The results of cluster analysis of gene expression by smRandom-seq are consistent with the morphologic features. Three subpopulations of *E. coli* display distinct gene expression patterns of SOS response and metabolic pathways, which might be interesting targets for antibiotic resistance research. By combining the high-sensitive cDNA generation chemistry with the popular droplet barcoding platform, we envision smRandom-seq will have a broad impact and facilitate discoveries in microbiology.

## Results

### Overview of the droplet-based smRandom-seq

The workflow of smRandom-seq is schematically shown in Fig. 1. First, bacteria were fixed overnight with ice-cold 4% paraformaldehyde (PFA) to crosslink the RNAs, DNAs, and proteins inside the bacteria. The fixed bacteria were then permeabilized to facilitate the following in situ reactions. Next, random primers with a GAT 3-letter PCR handle were added to capture total RNAs through multiple temperature cycling to enable maximum bindings of primers on each transcript inside bacteria. cDNAs were then converted in situ by reverse transcription reaction. Subsequently, poly(dA) tails were added to the 3' hydroxyl terminus of the cDNAs in situ by terminal transferase (TdT). Excessive primers, primer dimers, and leftover reagents were washed away by centrifugal washing after each step of in situ reactions. In our experimental pipeline, in situ reactions can be finished in two steps (~3 h). Before proceeding to microfluidic encapsulation, bacteria were imaged to confirm single bacterial morphology and manually counted under a microscope (Supplementary Fig. 1a). Each bacterium was encapsulated into a ~100-μm droplet with a poly(T) barcoded bead by using a modified microfluidic device (Supplementary Fig. 1b–d). To efficiently barcode the low content RNAs in single bacterium, we optimized the droplet barcoding platform (Supplementary Figs. 1b, c, 2a) based on our previous inDrop platform[20], with smaller barcoded beads (~40 μm) (Supplementary Fig. 2b), smaller droplets (~100 μm) (Supplementary Fig. 1d), a USER enzyme cutting strategy replacing the photocleaving strategy to release primers from barcode beads (Supplementary Fig. 2c), and a 3-step ligation reaction replacing the 2-step extension reaction for barcodes synthesis on the beads (Supplementary Fig. 2c). Shorter and cleaner barcode primers were synthesized by the ligation reaction (Supplementary Fig. 2d). During the barcoding reaction in droplets, the poly(T) primers were released from barcoded beads by the USER enzyme. Simultaneously, the cDNAs were released from bacteria by the RNase H enzyme. Then poly(T) primers bind with

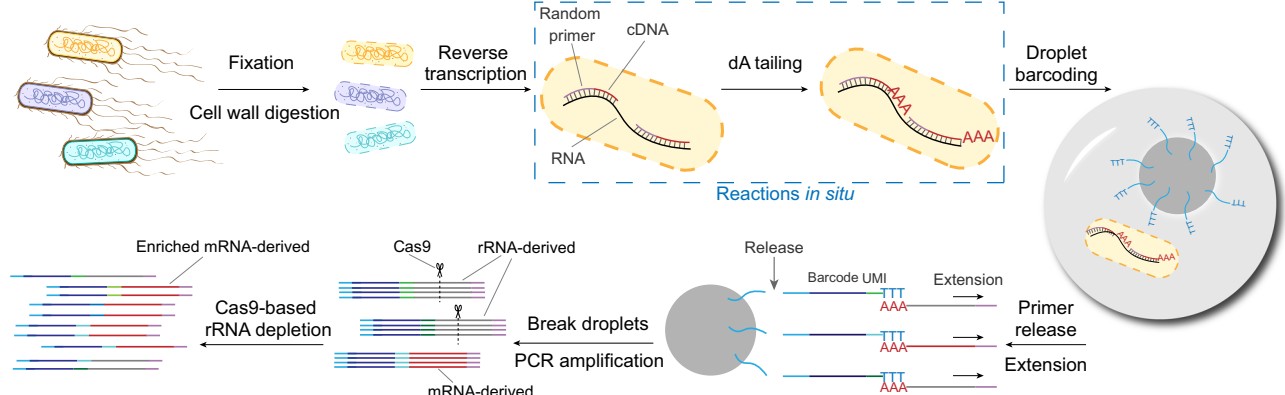

**Fig. 1 | Overview of the droplet-based smRandom-seq.** The workflow of smRandom-seq for microbe samples includes fixation, cell wall digestion, reverse transcription, dA tailing, droplet barcoding, primers releasing and extension, droplets breaking and PCR amplification, and Cas9-based rRNA depletion and sequencing. Blue dashed box: the two in situ reactions, including reverse transcription and dA tailing. AAA: dA tail in the 3' of cDNA, TTT: poly(dT) in the barcoded primers.

the poly(A) tails on the end of the cDNAs and extend to add a specific barcode to the cDNAs in each droplet and a unique molecular identifier (UMI) to each cDNA. These optimizations increased the barcoding efficiency for a single bacterium and reduced primer-dimers during the barcoding reaction in droplets. The diversity of barcode beads in this study was 442,368 (96 × 96 × 48). The throughput of current smRandom-seq is estimated from the Poisson distribution described by Zilionis et al.[22]. About 10,000 cells can be processed per experiment using the current smRandom-seq.

After barcoding, we broke the droplets and amplified the barcoded cDNAs. CRISPR-based rRNA depletion was performed on the cDNA library before next-generation sequencing. As a result of the optimized first- and second-strand cDNAs synthesis, the cDNA library prepared by our method ranged from 200 to 500 bp (Supplementary Fig. 1e), an optimal size range with no need to fragment library for next-generation sequencing platforms. After cutting the total length of the

adaptors (123 nt), the actual length of the mRNA-derived insert in libraries of smRandom-seq is about 100–400 bp.

## Validation of smRandom-seq method using reference bacteria species

We first performed the smRandom-seq assay on a two-species-mixing experiment of *E. coli* (Gram-negative bacteria) and *B. subtilis* (Gram-positive bacteria) to assess the purity of libraries (Fig. 2a). After sequencing and data processing, we used the molecular barcodes for unique cDNAs (UMI) to quantify smRandom-seq. A minor inter-species doublet rate (1.6%) and a very high species specificity of UMI (*B. subtilis*: 99.6%, *E. coli*: 98.4%) were calculated from the alignment results (Supplementary Fig. 3a, Fig. 2b). We detected a median of 6564 and 1785 UMI counts per cell for *B. subtilis* and *E. coli*, as well as a median of 1249 and 429 detected genes per cell, respectively (Supplementary Fig. 3b–e). There were 5 overlapped gene IDs between the top 10

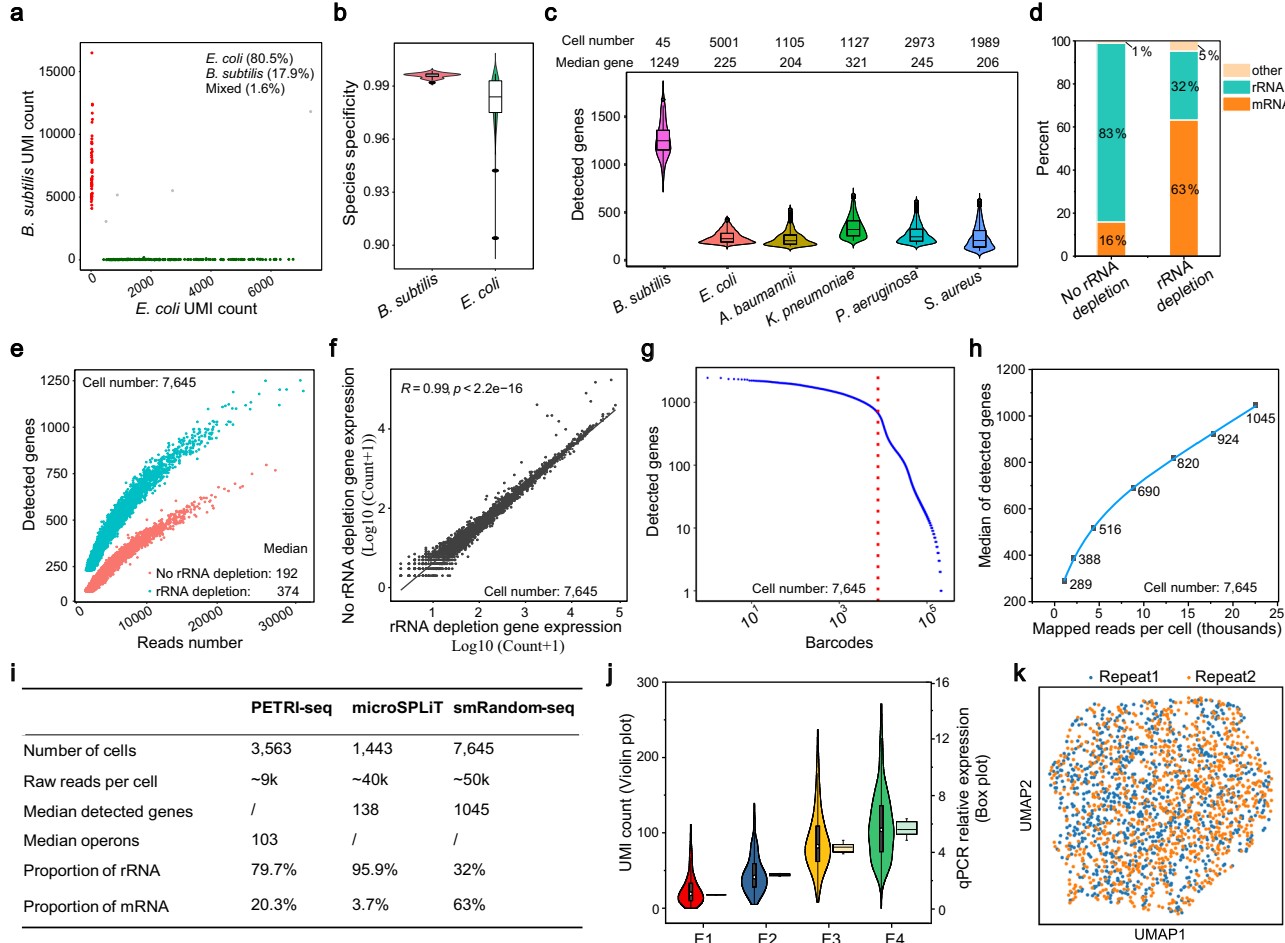

**Fig. 2 | Validation of smRandom-seq using reference bacteria samples. a** Scatter plot of UMI counts per cell barcode in the mixture. *E. coli* number: 202. *B. subtilis* number: 45. Mixed cell number: 4. **b** Species specificity of UMI in the mixture. *E. coli* number: 202. *B. subtilis* number: 45. Data in the box plot in **b**, **c**, **j** corresponded to the first (lower hinges) quartiles, third quartiles (upper hinges), and median (center). **c** Distribution of detected genes of different bacterial species. The identified cell numbers were as follows: 45 *B. subtilis*, 5001 *E. coli*, 1105 *A. baumannii*, 1127 *K. pneumoniae*, 2973 *P. aeruginosa*, and 1989 *S. aureus*. **d** Representative proportions of transcript categories of *E. coli* samples with and without rRNA depletion. The cell number of the *E. coli* samples in **d**–**f** was 7645. **e** Scatter plot showing detected genes and the number of reads per barcode of *E. coli* samples with (blue) and without (red) rRNA depletion. **f** Correlation of gene expression (Log10(Counts + 1)) of *E. coli* samples with and without rRNA depletion. The *E. coli* sample with rRNA depletion was sequenced more deeply for the next performance evaluations. **g** The

barcode rank plot of log10(total number of detected genes per barcode) versus log10 (Barcode number) for each possible barcode. The knee point (red dashed line) indicated the threshold for putative *E. coli* (cell number: 7645). **h** Saturation curve for the rRNA depleted and deeper sequenced *E. coli* sample (Cell number: 7645). **i** Comparison of performance with other two reported high-throughput single bacterium RNA-seq methods (PETRI-seq:[15] exponential-phase *E. coli* dataset from experiment 2.01, microSPLiT[16]: *E. coli* dataset form the *E. coli* and *B. subtilis* species-mixing experiment). **j** Comparison of qPCR relative expression (*n* = 3 biologically independent samples) and UMI count of *E. coli* samples (E1 *n* = 252 cells, E2 *n* = 254 cells, E3 *n* = 254 cells, E4 *n* = 253 cells) with a progressive increase in expression of GFP. E1–E4 represents the GFP-*E. coli* samples induced with 0, 6.25, 25, and 100 mM propionate. **k** UMAP projection of two repeated *E. coli* samples (Repeat 1 and Repeat 2) applied with smRandom-seq separately. Source data are provided as a Source Data file.

highest expression genes in the *B. subtilis* datasets of the species mixture by smRandom-seq and the public bulk RNA-seq dataset[23] (Supplementary Fig. 3f). We also performed a three-species mixture experiment of *A.baumannii*, *K. pneumoniae*, and *E. coli* by smRandom-seq (Supplementary Fig. 4a). All these species clustered separately, resulting from uniform manifold approximation and projection (UMAP)'s reduction (Supplementary Fig. 4b). The inter-species double rate of the three-species mixture experiment was 2.8%. However, the current smRandom-seq used in species-mixing experiments showed bias in cell number, UMI count, and detected genes due to variations in bacterial size, RNA content, and cell wall composition. We next tested other common clinical pathogenic bacteria (including Gram-negative bacteria: *E. coli*, *A. baumannii*, *K. pneumoniae*, *P. aeruginosa*, and Gram-positive bacteria: *S. aureus*) to confirm the general applicability of smRandom-seq. smRandom-seq method displayed desirable performance in these different bacterial species. A median of 225 genes and 428 UMIs was detected in 5001 *E. coli*; a median of 204 genes and 307 UMIs was detected in 1105 *A. baumannii*; a median of 321 genes and 610 UMIs was detected in 1127 *K. pneumoniae*; a median of 245 genes and 324 UMIs was detected in 2973 *P. aeruginosa*; a median of 206 genes and 393 UMIs was detected in 1989 *S. aureus* (Fig. 2c, Supplementary Fig. 5a). As shown on the UMAP plot, these bacterial species are also clustered separately (Supplementary Fig. 5b). As expected, smRandom-seq captured all biotypes of bacterial RNA, including mRNA, rRNA, tRNA, and ncRNA (Source data Fig. 2d). However, the mapped reads of bacterial libraries were dominated by rRNA (~80%) (Fig. 2d). To reduce the sequencing cost, we applied depletion of abundant sequences by hybridization (DASH) to deplete rRNA-derived cDNAs by CRISPR/Cas9[24]. We designed single guide RNAs (sgRNA) according to the *E. coli* rRNA sequencing results obtained above and optimized experimental conditions as described[24,25]. The rRNA-depletion treatment reduced the rRNA proportion from 83% to 32% (Fig. 2d). Correspondingly, mRNA proportion was multiplied four times (16–63%) (Fig. 2d). As a result, the sequencing sensitivity of smRandom-seq was increased by rRNA-depletion (Fig. 2e, Supplementary Fig. 6a). The medians of UMI count and detected genes of *E. coli* with rRNA depletion were 685 and 374, respectively. While the medians of UMI count and detected genes of *E. coli* without rRNA depletion were 332 and 192, respectively. CRISPR-based rRNA depletion did not change the gene expression pattern compared with the control ($R = 0.99$, $p < 2.2e{-}16$) (Fig. 2f). We also performed the Cas9-based rRNA depletion in other bacterial species, including *A.baumannii*, *K. pneumoniae*, *E. coli*, and *S. aureus*, and a mixed population of *A.baumannii*, *K. pneumoniae*, and *E. coli*. The rRNA-depletion treatment reduced the rRNA proportion from 71% to 10%, 88% to 29%, 74% to 4%, 91% to 45%, 83% to 5%, respectively (Supplementary Fig. 6b–f), which proved the broad applicability of our method. This rRNA-depleted *E. coli* library was further sequenced and displayed an obvious drop-off point (red dashed line) determining the threshold for actual bacteria with noise (Fig. 2g). Saturation analysis showed that the median of detected genes per bacterium continued increasing after 10k uniquely aligned reads per bacterium (Fig. 2h). The median of detected genes per bacterium could achieve ~1000 at the sequencing depth of ~20k mapped reads per bacterium (Fig. 2h). The performance of our smRandom-seq method is well above the published high-throughput single-bacterium RNA-seq methods, PETRI-seq[15] and microSPLiT[16] (Fig. 2i). The average coverage across all transcripts at each percentile of length gradually decreases from 5′ to 3′ end (Supplementary Fig. 7), which can be explained by the 5′–3′ direction of cDNA synthesis during RNA capturing.

To evaluate the accuracy of gene expressions by smRandom-seq, green fluorescence protein (GFP), as a quantitative reporter, was induced by propionate in *E. coli* with a GFP plasmid (E1–E4 samples: 0, 6.25, 25, and 100 mM propionate, respectively). GFP fluorescence increased, and Ct values of the GFP coding gene in qPCR measurement

decreased as the amount of propionate increased (Supplementary Fig. 8a, b). Consistent with the GFP fluorescence values and GFP mRNA expression levels, the median UMI counts of the GFP gene detected by smRandom-seq also increased along with the increased amount of propionate and correlated well with the relative GFP gene expression levels measured by fluorescent measurement ($R = 0.97$, $p = 0.033$) and qPCR ($R = 1$, $p = 0.0018$) (Fig. 2j, Supplementary Fig. 8c, d), suggesting that the gene expression by smRandom-seq is consistent with that by qPCR.

To prove the technical reproducibility of smRandom-seq, two repeated exponential-phase *E. coli* samples from the same culture tube were applied with smRandom-seq separately (Supplementary Fig. 9a). These two independent results displayed proper alignment visualizing with UMAP dimensionality reduction (Fig. 2k) and had a high correlation on gene expression ($R = 0.95$, $p < 2.2e{-}16$) (Supplementary Fig. 9b). These results verified that smRandom-seq could efficiently and accurately capture total RNAs in a single bacterium and is applicable to multiple bacteria species.

## smRandom-seq revealed transcriptome changes of single *E. coli* upon antibiotic stress

smRandom-seq was next applied to study the transcriptome programs in single *E. coli* under antibiotic stress. We chose ciprofloxacin (CIP), a fluoroquinolone antibiotic commonly used to treat *E. coli* infections. CIP induces antibacterial activity primarily via induction of double-stranded DNA breaks, as well as induction of oxidative stress[26]. Exponential stage *E. coli* was treated with a 15 µg/mL concentration of CIP, which could induce a decline of the growth curve upon antibiotic stress (Supplementary Fig. 10). The survived *E. coli* were sampled after 0, 1, 2, and 4 h, and applied with smRandom-seq, respectively (Fig. 3a). The morphology of bacteria in each group was homogeneous (Fig. 3b). The UMI count and detected genes in these CIP-treated *E. coli* samples gradually decreased after 1, 2, and 4 h (Supplementary Fig. 11a, b). There were 5 overlapped gene IDs among the top 10 highest expression genes in the *E. coli* datasets of the CIP-T0 sample, the species-mixture by smRandom-seq, and the public bulk RNA-seq dataset[27] (Supplementary Fig. 11c). Unsupervised graph-based clustering resulted in four distinct clusters, while the gene expression in each cluster was continuous (Fig. 3c), which is consistent with the morphologic feature. The strongly decreased expression of *ompF* was consistent with the reported CIP resistance mechanism through the decreased amount of porin outer membrane protein OmpF[28] (Fig. 3d). Gene expressions of other two outer membrane protein-encoding genes, *tsx* and *lamB*, were also reduced (Fig. 3d). We ranked marker genes in each group and identified the top 5 differentially expressed genes (DEGs) to characterize the transcriptional profile of each *E. coli* sample (Fig. 3e). Among the top 5 DEGs for 2 h time point upon CIP treatment, *recA* was reported to promote DNA repair and recombination, contributing the ciprofloxacin resistance[29], and *tisB* was reported to be a persister gene that induces reversible dormancy by shutting down cell metabolism[30]. Consistently, the top 20 DEGs of group CIP 2 h were enriched in the cellular response to DNA damage stimulus (Fig. 3f). We next analyzed the expression patterns of genes involved in SOS response and metabolic pathways (Fig. 3g, h). Genes involved in SOS response displayed an acute increase upon CIP treatment, especially at 1 h, which is consistent with the fact that fluoroquinolones are potent inducers of the SOS response[31] (Fig. 3f). The inhibited reactive oxygen species (ROS) degradation genes upon CIP treatment were consistent with reported ROS accumulation induced by CIP[32] (Fig. 3g). The expression levels of metabolic genes were significantly reduced with treatment time (Fig. 3h), suggesting a metabolic shutdown in CIP-treated *E. coli*, avoiding metabolic toxicity, and minimizing drug lethality[33], which is also consistent with the reduced UMI count and detected genes in those samples showed above (Supplementary Fig. 11a, b). These results suggested that smRandom-seq could sensitively capture the

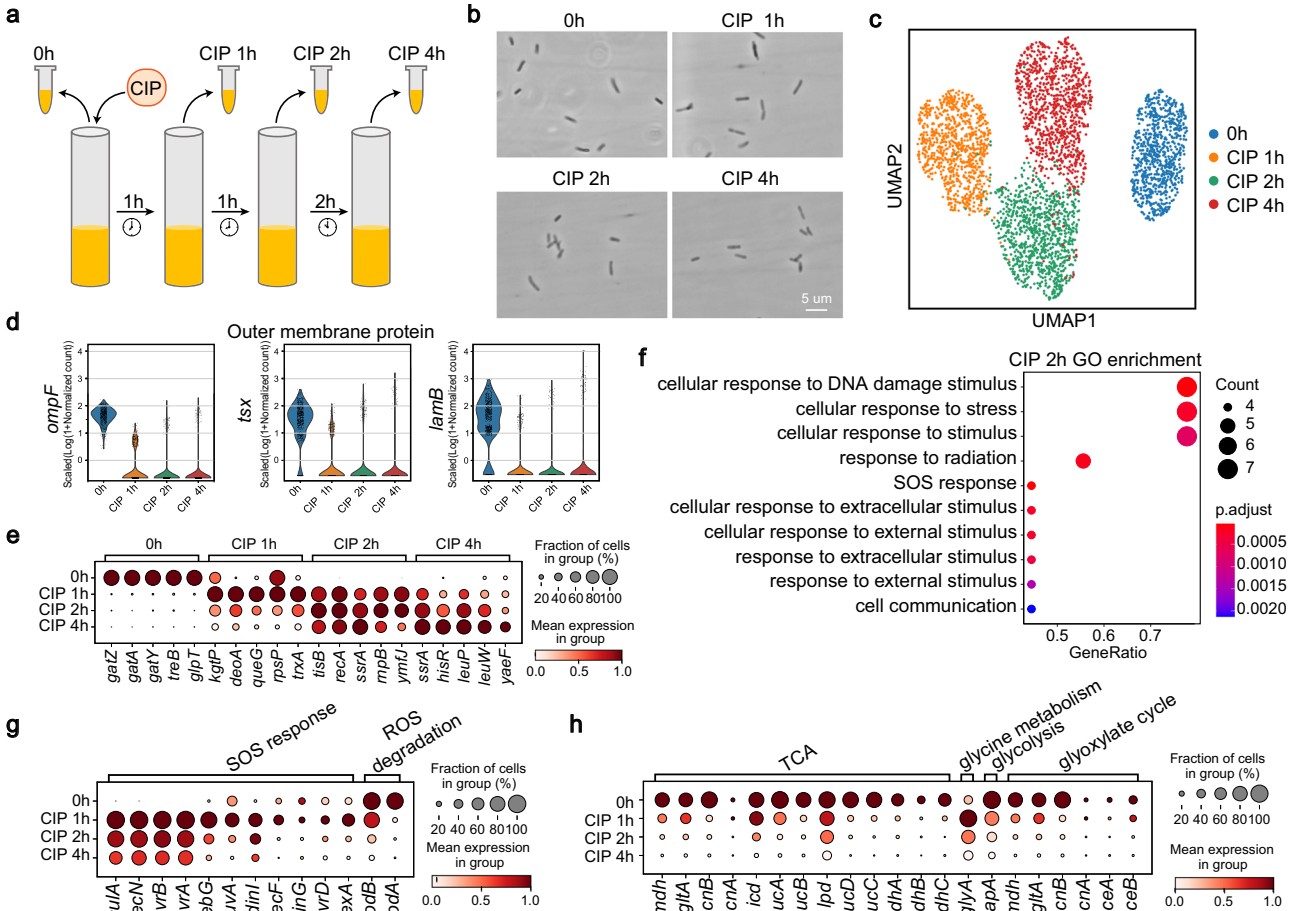

**Fig. 3 | smRandom-seq captured transcriptome changes of single *E. coli* upon antibiotic stress. a** Experimental design for CIP (ciprofloxacin, 15 µg/mL)-treated *E. coli* samples. **b** Images of *E. coli* treated with CIP for 0, 1, 2, and 4 h. *n* = 4 independent experiments. **c** UMAP projection of all the bacteria collected at the different time points, based on their gene expression colored by time point. **d** Violin plot showed the expression levels of outer membrane protein-encoding genes (*ompF*, *tsx*, and *lamB*) in samples. 0 h *n* = 771 cells, CIP 1 h *n* = 850 cells, CIP 2 h *n* = 950 cells, CIP 4 h *n* = 949 cells. The expression levels of each gene in each cell were normalized, log-transformed (log(1 + *x*)), and scaled data. The violin plots showed the kernel density estimate of the underlying data. **e** Mean expression levels of top 5 DEGs among different samples. The color of each dot represents the mean expression within each sample and the size of each dot represents the

fraction of cells expressing the DEGs in different samples. **f** Bubble plots of GO enrichment analysis of top 20 DEGs in CIP 2 h sample. Gene count: the number of DEGs enriched in a GO term. Gene ratio: the number of observed DEGs divided by the number of expected genes from each GO term. The cutoff value of the *p*-value of GO enrichment was 0.05 and the cutoff value of the *q*-value of GO enrichment was 0.05. **g**, **h** Mean expression levels of genes involved in SOS-response and reactive oxygen species (ROS) degradation (**g**) and different metabolic pathways (**h**) in different time point samples. The color of each dot represents the mean expression within each sample and the size of each dot represents the fraction of cells expressing the specific genes in different samples. TCA tricarboxylic acid cycle. Source data are provided as a Source Data file.

transcriptome changes of *E. coli* upon antibiotic stress at the single-bacterium level.

### smRandom-seq discovered antibiotic-resistant subpopulations in *E. coli* upon antibiotic stress

To further characterize the heterogeneous response of bacteria upon another antibiotic that is widely used in human and livestock *E. coli* infection, we first treated exponential-stage *E. coli* with ampicillin (AMP), a semi-synthetic β-lactam antibiotic, at a concentration (5 µg/mL) inducing apparent bacteria death according to the growth curve (Supplementary Fig. 12a). All the *E. coli* showed a clear elongation after 4 h treatment of AMP at 5 µg/mL (Supplementary Fig. 12b). We next increased the concentration of AMP (12.5 µg/mL) and found that the morphology of the survived *E. coli* displayed an increasing heterogeneity towards the 4-h time point (Fig. 4a). To profile the transcriptome of these survived subpopulations, we performed smRandom-seq on these *E. coli* samples treated with 12.5 µg/mL AMP for 0, 1, 2, and 4 h. The UMI count and detected genes of AMP-treated *E. coli* decreased after 2 and 4 h (Supplementary Fig. 13a, b). The

separated clusters (Fig. 4b) and the top 10 DEGs (Supplementary Fig. 14a) of the four-time points suggested that AMP treatment induced significant changes in gene expression patterns over time. For example, the DNA damage-related genes, *recA* and *sulA*, were significantly raised at the 4 h time point (Supplementary Fig. 14a), which is consistent with the increased proportions of dead cells and the decreased cell viabilities after AMP treatment (Supplementary Fig. 15a, b). Furthermore, different subpopulations of these AMP-treated samples were algorithmically clustered using the Leiden algorithm (Fig. 4c). Consistent with the bacterial morphology heterogeneity, gene expression patterns of these 12.5 µg/mL AMP-treated bacteria showed distinct subclusters at 2 and 4 h time points as visualized by a UMAP dimensionality reduction (Fig. 4c). The subclusters mainly from 2 h (subclusters 2, 5, and 8) and 4 h (subclusters 6, 9, and 10) time points showed low correlations of gene expression patterns (Fig. 4d), which suggested that the gene expression patterns diverge as the bacteria were exposed to AMP over time. Therefore, we ranked genes for characterizing these subclusters and identified the top 5 DEGs to further understand the heterogeneity of *E. coli* upon AMP treatment

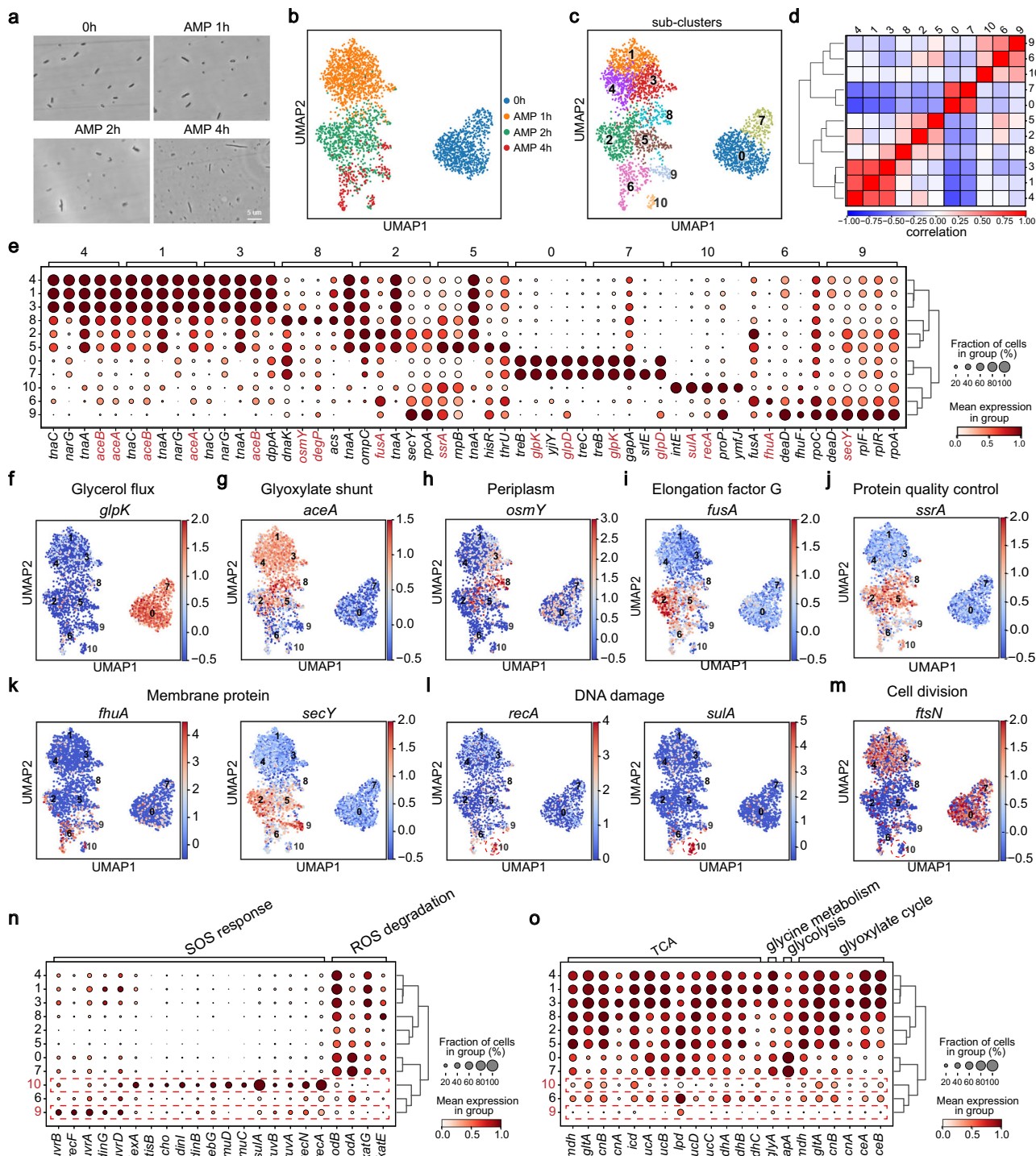

**Fig. 4 | smRandom-seq identified subpopulations reacting differently to ampicillin. a** Images of *E. coli* treated with AMP (ampicillin, 12.5 μg/mL) for 0, 1, 2, and 4 h. *n* = 4 independent experiments. **b, c** UMAP projection of all the cells collected at the different time points, based on their gene expression colored by time points (**b**) or sub-clusters (**c**). **d** The correlation matrix showed the correlation coefficients between sub-clusters. **e** The mean expression levels of the top 5 DEGs among different subclusters. The color of each dot represents the mean expression within each subcluster and the size of each dot represents the fraction of cells expressing the DEGs in different subclusters. **f–l** Expression levels of the glycerol flux-related marker gene of subclusters 0 and 7 (*glpK*) (**f**), the glyoxylates shunt-related marker gene of subclusters 1, 3 and 4 (*aceA*) (**g**), the periplasm-related marker gene of subcluster 8 (*osmY*) (**h**), the elongation factor G related marker gene of subcluster 2 (*fusA*) (**i**), the protein quality control related marker gene of subcluster 5 (*ssrA*) (**j**), the membrane related marker genes of subclusters 6 and 9 (*fhuA*, *secY*, respectively) (**k**), the two DNA damage related marker genes of subcluster 10

(*recA*, *sulA*) (**l**) in each cell overlaid on the UMAP plot. The expression levels of each gene in each cell were normalized, log-transformed (log(1 + *x*)), and scaled (zero mean and unit variance) data. Dashed red circle: the distinct sub-cluster 10.
**m** Expression of cell division-related gene (*ftsN*) overlaid on the UMAP plot. Dashed red circle: the distinct sub-cluster 10. **n** Mean expression levels of genes involved in SOS-response and ROS degradation in different sub-clusters. Dashed red rectangle: the distinct gene expression patterns of sub-clusters 9 and 10, which are associated with a 4 h time point. **o** Mean expression levels of genes involved in different metabolic pathways in different subclusters. The color of each dot represents the mean expression within each subcluster and the size of each dot represents the fraction of cells expressing the genes involved in different metabolic pathways in different subclusters. TCA tricarboxylic acid cycle. Dashed red rectangle: the distinct gene expression patterns of sub-clusters 9 and 10, which are associated with 4 h time point.

(Fig. 4e). We selected these antibiotic response-related DEGs (including *aceA*, *aceB*, *osmY*, *degP*, *fusA*, *ssrA*, *glpD*, *glpK*, *sulA*, *recA*, *fhuA*, and *secY*) representing each subcluster (Fig. 4e). Further, we investigated the feature expression in each bacterium in low-dimensional space (UMAP) (Fig. 4f–l, Supplementary Fig. 14b–d). The aerobic fermentation of glycerol-associated genes (*glpD* and *glpK*) highly and specifically expressed at 0 h time points (subclusters 0 and 7) (Fig. 4f, Supplementary Fig. 14b), followed by general high expression of the genes (*aceA* and *aceB*) that encode enzymes of the glyoxylate shunt at the 1 h time point (subclusters 1, 3 and 4) (Fig. 4g, Supplementary Fig. 14c). These results suggested that the aerobic fermentation of *E. coli* run at high growth rates at the 1 h time point. As time went on, *aceA* and *aceB* showed distinct expression patterns among subclusters, especially the highest upregulation in subcluster 8 of the 2 h time point (Fig. 4g, Supplementary Fig. 14c). Subcluster 8 displayed markable expressions of genes (*osmY* and *degP*) related with periplasm (Fig. 4h, Supplementary Fig. 14d), which are critical for bacterial stress resistance[34,35]. Subcluster 2 displayed a markable expression of *fusA* (Fig. 4i), encoding elongation factor G, a critical target during oxidative damage to the translation system of *E. coli*[36]. Subcluster 5 was active in protein-quality control of translation with a high expression of *ssrA* (Fig. 4j) that mediates degradation of aberrant, unfinished proteins when *E. coli* ribosomes stall during translation[37]. These results suggested that subpopulations of *E. coli* undergo different degrees of membrane integrity and translation damage upon 2-hour AMP treatment. Subclusters 6 displayed a high expression level of *fhuA* which encodes *E. coli* ferric hydroxamate protein uptake component A, and subclusters 9 displayed a high expression level of *secY* which encodes nonspecific sugar transporters in *E. coli*, respectively (Fig. 4k). The membrane protein-encoding gene, *secY*, also had high expression level in subcluster 2 of 2-h time point, suggesting that subcluster 9 might be developed from subcluster 2. Gene Ontology (GO) enrichment analysis revealed that the marker genes (*rpoC*, *rplF*, *rplR*, *rpoA*) of subclusters 6 and 9 also enriched in ribosome assembly (Fig. 4e, Supplementary Fig. 14e, f). Severe DNA damage of bacteria in subcluster 10 could be inferred, as *recA* and *sulA* both showed an apparent enrichment in subcluster 10, where the former is essential in DNA repair[38] and the latter is inducible by DNA damaging agents[39] (Fig. 4l). The top 20 DEGs of subcluster 10 were mainly enriched in cellular response to stimulus and SOS response (Supplementary Fig. 14g), which is supported by the fact that both *recA*[40] and *sulA*[39] are SOS response regulators. Consistent with that SOS response aids bacterial propagation by inhibiting cell division during repair of DNA damage[40], cell division gene *ftsN* was inhibited at the corresponding point on the UMAP plot (Fig. 4m). More SOS response- and ROS degradation-related genes displayed different expression patterns in subclusters 6, 9, and 10, which were associated with the 4-h time point (Fig. 4n). In addition, GO enrichment analysis revealed enrichment in metabolic process pathways in subclusters 6 and 9 (Supplementary Fig. 14e, g). The expression levels of metabolic-related genes were increased after 1 h of antibiotic treatment and then decreased at 2- and 4-h time points (Supplementary Fig. 14h), which is consistent with the trend of the average respiratory activity of *E. coli* first increasing and then decreasing after antibiotic treatment (Supplementary Fig. 16a, b). At the 4-h time point, we observed strong downregulation of metabolic-related genes in sub-clusters 9 and 10 but upregulation in sub-cluster 6 (Fig. 4o). These findings suggest a low metabolic activity that can generate ROS, which is consistent with the low expression of ROS degradation genes and high expression of SOS response genes in sub-clusters 9 and 10 (Fig. 4n). These findings were also consistent with the observation that a sub-population of bacteria at the 4-h time point displayed high respiratory activity, as evidenced by high levels of CTC fluorescence intensity (Supplementary Fig. 16b). These results showed that *E. coli* population responded differently to high-concentration AMP. This heterogeneity is ignored by traditional bulk RNA-seq methods but well characterized by smRandom-seq.

## Discussion

In this study, we developed a droplet-based, high-throughput, and high-sensitivity single-microbe RNA-seq method, named smRandom-seq, with the pipeline of bacteria preparation, reactions in situ, droplet barcoding, and library preparation. We provided sufficient and systematic evidence to prove the efficiency and technical reproducibility of smRandom-seq and validated the performance of smRandom-seq by practical application.

We used a GAT three-letter code primer with seven random nucleotides as the RT primer, which can evenly and multiply hybridize to total RNA templates at low temperature[41]. The above results proved that this adapted RT primer is efficient for the reverse transcription of fixed bacterial RNA with a complex structure inside a single bacterium. Otherwise, the current popular scRNA-seq platform 10X Genomics[7], as well as the single microbe RNA-seq method PETRI-seq[15] and microSPLiT[16], used a template-switching strategy for the second strand cDNA synthesis. The template-switching strategy could be affected by the complex structure of RNAs and fail to capture interrupted cDNAs[42]. Thus, we adapted a dA tailing strategy for efficient second-strand cDNA synthesis, as non-template dA can be added to the 3′ OH terminus of all cDNA fragments, including interrupted products, by terminal deoxynucleotidyl transferase. Unlike the multiple split-pool ligation-based barcoding in PETRI-seq[15] and microSPLiT[16], the barcoding strategy of smRandom-seq uses an efficient and stable droplet microfluidics platform to co-encapsulate single bacterium with uniquely barcoded beads. Recently, three bacterial single-cell RNA-seq methods, ProBac-seq[43], M3-Seq[44], and BacDrop[45] were reported using the commercially 10X Genomics microfluidic single-cell sequencing platform. To efficiently barcode the low-content RNAs in the single bacterium, we optimized our previous droplet barcoding platform designed for single-cell barcoding[20], by generating smaller barcoded beads and droplets and modifying the primers releasing and the barcodes synthesis strategy. These optimized strategies in the smRandom-seq method come out high-quality cDNA library with 200–500 bp size from a single bacterium, which is not needed to fragment but is just suitable for the next generation sequencing (NGS). A pre-index strategy could be easily combined with the smRandom-seq by adding indices to RT primers according to the published scifi-RNA-seq[46], where the bacteria were split into different tubes for RT with pre-indexed random primers, and then those pre-indexed samples could be combined into one sample for the following barcoding procedures. These double-end barcoded cDNAs need to be directly paired-end sequenced without fragmentation. Pre-indexed smRandom-seq would not only extend the throughput of our method but also reduce spurious biological conclusions of doublets arising during microfluidics encapsulation. In addition, this pre-indexed smRandom-seq will be suitable for processing multiple samples simultaneously.

The efficiency of smRandom-seq is mainly affected by an overwhelming quantity of rRNAs (>80% of mapped reads) which are usually not of interest. To avoid the cost of sequencing non-messenger transcripts in such large-scale bacterial transcriptomics studies, here we incorporated an rRNA depletion step at the library level according to the reported DASH method[24,25]. Our results indicated that this CRISPR-based technique markedly improved the performance of smRandom-seq on single *E. coli* RNA sequencing with a median of ~1000 genes detected per bacterium by efficiently removing rRNA-derived cDNAs. Recently, Homberger et al. also posted successful Cas9-based rRNA depletion for the libraries of *Salmonella* by a low-throughput bacterial sing cell RNA-seq method[47]. Wang et al. reported another post-hoc rRNA depletion scheme for M3-Seq[44], which used a duplex-specific nuclease to remove the rRNA hybridized with DNA probes targeting rRNAs of specific bacterial species. BacDrop depleted rRNAs in cells prior to reverse transcription also using the duplex-specific nuclease[45]. In order to apply smRandom-seq to other microbial species and even

more complex natural communities, a universal microbiome rRNA depletion method[48] is needed to be incorporated into smRandom-seq.

We also applied smRandom-seq to profile transcriptome changes of single *E. coli* under antibiotic treatment. These results found a few antibiotic-resistant subpopulations in antibiotic-treated *E. coli* displaying distinct gene expression patterns of SOS response and metabolic pathways. Our study provides a way to tease out how individual bacterium adapts and interacts with each other under environmental stress. We can then efficiently predict which subpopulations will evolve resistance and survive to impact health, as well as figure out the mechanisms of bacterial resistance and persistence, which would be valuable for precision diagnosis and treatment of bacterial infection. Moreover, combined with the Combi-Seq[49] which uses a combinatorial DNA barcoding approach to encode treatment conditions in droplets, smRandom-seq would probably allow monitoring the bacterial transcriptomic changes at a single bacterium level for high-throughput drug screening. We also anticipate that smRandom-seq would be broadly applied in revealing heterogeneous transcriptome profiles of microbiota, identifying rare species, and deciphering cross-species interactions in more varied environments, such as gut microbiota and soil microbiota.

## Methods

### Bacterial culture
*Escherichia coli* BW25113 (*E. coli*), *Bacillus subtilis* NCBI3610 (*B. subtilis*), *Acinetobacter baumannii* ATCC17978 (*A. baumannii*), *Klebsiella pneumoniae* XH209 (*K. pneumoniae*), *Pseudomonas aeruginosa* PAO1 (*P. aeruginosa*) and *Staphylococcus aureus* subsp. *Aureus* SA268 (*S. aureus*) was provided by Sir Run Run Shaw Hospital, Zhejiang University School of Medicine. Overnight cultured bacteria were inoculated into fresh LB Lennox medium at 1:1000, respectively, and grown at 37 °C with shaking at 250 rpm.

For the two-species-mixing experiment, cultures of *E. coli* (Gram-negative bacteria) and *B. subtilis* (Gram-positive bacteria) were sampled upon reaching the OD600 ~ 0.5, immediately centrifuged at 4 °C, 2000×*g* for 2 min, next washed twice by PBS, and mixed before applied with smRandom-seq.

For the three-species-mixing experiment, cultures of *E. coli*, *A. baumannii* and *K. pneumoniae* were sampled upon reaching the OD600 ~ 0.5, immediately centrifuged at 4 °C, 2000×*g* for 2 min, next washed twice by PBS, and mixed before applied with smRandom-seq.

For single-species experiments, cultures of Gram-negative bacteria: *E. coli*, *B. subtilis*, *A. baumannii*, *K. pneumoniae*, *P. aeruginosa*, and Gram-positive bacteria: *S. aureus* were sampled upon reaching the OD600 ~ 0.5, immediately centrifuged at 4 °C, 2000×*g* for 2 min, next washed twice by PBS, and applied with smRandom-seq, respectively.

For the GFP quantitation experiment, *E. coli* harboring pPro24-gfp[50] (GFP-*E. coli*) was inoculated into fresh LB medium supplemented with ampicillin (100 µg/mL) and grown at 37 °C. When the OD600 reached 0.5, GFP-*E. coli* samples were induced with 0, 6.25, 25, and 100 mM propionate (E1–E4 samples, respectively), respectively. GFP fluorescence was measured in a SpectraFluor Plus plate reader (Tecan) using an excitation wavelength of 405 nm and an emission wavelength of 535 nm. GFP fluorescence was normalized for cell density (GFP fluorescence/OD600).

The bactericidal effect experiments for CIP and AMP were prepared by inoculating overnight cultured *E. coli* into fresh LB medium and growing until OD600 ~ 0.1 was reached. CIP was added to a final concentration of 0, 6.25, 12.5, 25, and 50 µg/mL, respectively. AMP was added to a final concentration of 0 and 5 µg/mL, respectively. The optical density at 600 nm (OD600) of bacterial cell suspensions at different stages was measured with an xMark™ microplate absorbance spectrophotometer (Bio-Rad).

The antibiotic-treated samples for smRandom-seq were prepared by inoculating overnight cultured *E. coli* into fresh LB medium and growing until OD 0.5 was reached. 0 h sample was taken, and 15 µg/mL CIP, 5 µg/mL AMP, or 12.5 µg/mL AMP was added to the rest *E. coli*, respectively. The following samples were taken from the culture at 1, 2, and 4 h time points, respectively. Samples were centrifuged at 4 °C, 2000×*g* for 2 min, washed twice with PBS, and applied with smRandom-seq, respectively.

### Fixation and permeabilization
Bacteria were fixed in 4% PFA (Beyotime Biotechnology, Cat. P0099) at 4 °C overnight, then washed and permeabilized by 0.04% Tween-20 (Sangon Biotech, Cat. 9005-64-5) in PBS. The cell wall was digested with lysozyme by incubating at 37 °C for 15 min. The lysozyme was ordered from Thermo Fisher (Cat. 90082). Following the cell wall digestion step, bacteria were immediately washed and resuspended in PBS with RNase inhibitor (Thermo Fisher Scientific, Cat. N8080119) for the next step.

### In situ reverse transcription
In situ, reverse transcription of bacteria was performed according to the procedures of the reverse transcription kit. The reverse transcription kit (Cat. R20114124), including reverse transcriptase (50 U/µL), 5× reverse transcription buffer, and dNTP Mix (100 mM), was ordered from M20 Genomics. The RT reaction mix (per 50-µL reaction) was prepared on ice, including 1–10 million bacteria in 27.5 µL PBS, 10 µL 5× reverse transcription buffer, 5 µL 10 µM random primer (sequence in Supplementary Table 1), 2.5 µL 100 mM dNTP, 2.5 µL RNase inhibitor, 2.5 µL reverse transcriptase (50 U/µL), and incubated with twelve cycles of multiple annealing ramping from 8 to 42 °C, followed by a 42 °C 30 min, in a thermal cycler. After reverse transcription, bacteria were washed with PBST (PBS with 0.05% Tween-20) five times.

### In situ dA tailing
For In situ dA tailing of bacteria, the following reaction mix was prepared, including a few million bacteria in 39 µL PBS, 5 µL 10X TdT buffer, 5 µL 2.5 mM CoCl₂, 0.5 µL 100 mM dATP, 0.5 µL TdT enzyme, and incubated at 37 °C for 30 min, and then washed with PBST three times. The TdT reaction kit (including TdT enzyme, 10X TdT buffer and CoCl₂) was ordered from New England Biolabs (Cat. M0315S).

### Microfluidic device fabrication
PDMS-based microfluidic devices were designed for hydrogel beads synthesis and single bacterium barcoding, and fabricated[22,51]. The channel depth of microfluidic devices for hydrogel beads synthesis is 30 µm (Supplementary Fig. 2a). The channel depth of microfluidic devices for cell encapsulation is 50 µm (Supplementary Fig. 1b). Molds for a microfluidic device were made using a photolithographic approach, consisting of centrifugally coating and modeling the SU-8. The silicon molds were cast with PDMS (Sylgard-184) to fabricate microfluidic devices.

### Microfluidic platform
The microfluidic platform for hydrogel beads synthesis and single bacterium barcoding was established[22,51,52], which includes microfluidic devices, two or three syringe pumps, some syringes (1 mL) and tubing, and an inverted bright-field microscope with a fast speed camera and a computer.

### Barcoded beads synthesis
The hydrogel barcoded beads for single bacterium barcoding were designed[22,51,52] (Supplementary Table 1) and customized by M20 Genomics company. Hydrogel beads were synthesized by the microfluidic emulsification and polymerization of an acrylamide-primer mix.

The acrylamide-primer mix contains acrylamide:bis-acrylamide solution (1×) (Invitrogen, Cat. AM9022), acrydite-modified oligonucleotides (50 μM) that is covalently incorporated into the hydrogel mesh, ammonium persulfate (APS, 10% wt/vol, Sangon Biotech Cat. A100486-0025) that initiates the free-radical polymerization, and Tris-buffered saline-EDTA-Triton (TBSET) buffer (1×). The photocleavable moiety contained in the acrydite-modified oligonucleotide on previously reported hydrogel beads[22] was replaced by a deoxy Uridine base in our customized protocol. A carrier oil–TEMED mix (1 mL:4 μL) was applied in microfluidic. The acrylamide-primer mix and carrier oil-TEMED mix were transferred to syringes, respectively. Then, the syringes were connected to corresponding inlets of the 30-μm-deep hydrogel bead synthesis device (Supplementary Fig. 2a) via tubing, respectively. The flow rates for the acrylamide-primer mix and carrier oil-TEMED mix were 900 and 1800 μL/h, respectively. The size of generated hydrogel beads should be 40 μm (Supplementary Fig. 2b). Next, the DNA primers on the generated hydrogel beads are barcoded using a combination of a split-and-pool method and a 3-step primer ligation reaction (Supplementary Fig. 2c), instead of the previous reported 2-step extension reaction[22]. The unique barcoded primers were provided in Supplementary Table 2. Hydrogel beads mix, including hydrogel bead, DNA ligase (350 U/mL, New England Biolabs, Cat. M0202S), dNTP (10 mM each), 1× isothermal amplification buffer (New England Biolabs, Cat. B0537S), nuclease-free water was prepared, then split into a round-bottom 96-well plate. The hydrogel beads mix in 96-well plates were mixed with 96 annealed unique barcode primers in 96-well plates, respectively, then incubated at 37 °C for 30 min. All hydrogel beads were combined in a single 50-mL tube and washed with STOP-25 buffer[22], then performed with the second (96 unique barcoded primers) and third (48 unique barcoded primers) split-and-pool rounds. For quality control of the generated hydrogel barcoded beads, barcoded primers were released by USER (New England Biolabs, Cat. M5505S) enzymatic digestion and analyzed with gel electrophoresis (Supplementary Fig. 2d). The highest-molecular-weight peak (that typically appears at ~100 bp) in the electropherogram represents the full-length barcoding primer (actual size 96 nt), and lower-molecular-weight peaks are synthesis intermediates (actual size 27, 44, 58 nt) (Supplementary Tables 1, 2). All the required reagents for hydrogel barcoded beads synthesis and the ready-to-use hydrogel barcoded beads can be ordered from the M20 Genomics company.

### Droplet barcoding

The modified droplet barcoding for a single bacterium was performed according to previous work[51,52]. The qualitied single bacteria were counted under an optical microscope and diluted with a 15% density gradient solution. Bacteria, 2X DNA extension reaction mix, and hydrogel barcoded beads were encapsulated using the microfluidic platform. The density gradient solution and 2X DNA extension reaction mix were ordered from M20 Genomics. The collected droplets were incubated at 37 °C for 1 h, 50 °C for 30 min, 60 °C for 30 min, and 75 °C for 20 min, and then broken by mixing with PFO buffer. The oil phase was discarded, and the aqueous phase containing cDNAs was purified by Ampure XP beads. The purified cDNAs were amplified by PCR with PCR1 and PCR2 primers (Supplementary Table 1), purified by Ampure XP beads, and quantified by Qubit 2.0 (Thermo Fisher Scientific).

### Ribosomal RNA depletion

We used Cas9-based depletion of abundant sequences by hybridization (DASH) treatment to remove the majority of ribosomal cDNA[24,25,53]. Target cDNA sequences from *E. coli* rRNA genes were identified, and sgRNAs were designed to target these sequences[25,53] (Supplementary Data 1). DNA templates for sgRNAs based on an optimized scaffold were made[24] and transcribed in vitro into sgRNAs using T7 RNA polymerase. The sgRNA pools were purified with VAHTS

RNA Clean Beads, aliquoted, and stored at −80 °C. T7 RNA polymerase was ordered from ThermoScientific (Cat. no. EP0111). VAHTS RNA Clean Beads were ordered from Vazyme (Cat. no. N412-01). Cas9 protein was ordered from NEW ENGLAND BIOLABS (Cat. no. M0386S). A reaction mixture of Cas9 and the sgRNA pool were preincubated at 37 °C for 15 min. Next, 1 ng purified PCR product was added to the Cas9–sgRNA complex. The ratio of the cDNA:Cas9 enzyme:sgRNA is 1:100:1000. The cDNA–Cas9–sgRNA complex incubating at 37 °C for 2 h. After the digestion, Cas9 was inactivated by treatment with 0.8 U (-20 μg) proteinase K for 15 min at 37 °C, followed by heat-inactivation (15 min at 95 °C). sgRNA was digested by RNase If incubating at 37 °C for 15 min and 70 °C for 15 min. Proteinase K (Cat. no. P8107S) and RNase If (Cat. no. M0243S) were ordered from NEW ENGLAND BIO-LABS. The resulting DASH-treated samples were PCR amplified for 10–20 cycles to select for non-ribosomal, undigested cDNAs and purified with 1X AMPure XP beads.

### Library preparation

In this study, we used VAHTS Universal DNA Library Prep Kit for Illumina V3 (Vazyme, Cat. no. ND607-03/04) to construct the library. The amplified and purified cDNAs were quantified by Qubit 2.0 and measured by Qsep100™ DNA Fragment Analyzer (Bioptic). Then the qualified cDNAs were performed with end-repair and adenylation. The reaction mixture, containing 50 ng fragmented DNA, end repair enzymes, end repair buffer, and nuclease-free water, was prepared, incubated at 30 °C for 30 min, and then inactivated at 65 °C for 30 min. Working adaptor and ligation enzymes were added to the finished end-prep reaction mixture and then incubated at 20 °C for 15 min. The ligated DNA was purified and selected with AMPure XP beads. Then, the library was amplified by PCR and purified with AMPure XP beads. The final cDNA library was quantified by Qubit 2.0 and measured with Qsep100™ DNA Fragment Analyzer. Finally, the qualified library was sequenced by the NovaSeq 6000 and S4 Reagent Kit with paired-end reads of 150.

### Processing of sequencing data

The raw sequencing data were trimmed using Cutadapt (version 3.7)[54]. Primer sequences were removed, and extra bases generated by the dA-tailing step were trimmed. For each Read1, UMI (8 nts) and cell-specific barcode (30 nts) were extracted. Sequenced barcodes that can be uniquely assigned to an accepted barcode with a Hamming distance of 2 nt or less were merged. Read2 was mapped to the genome of the species of interest using STAR (2.7.10a)[55] with default parameters. Only uniquely mapped reads were used for downstream analyses. The GenBank assembly accession of reference genomes and annotation files used in this study are as follows: GCF_902728005.1 for *A. baumannii*; GCF_000750555.1 for *E. coli*, GCF_000775955.1 for *K. pneumoniae*, GCA_000006765 for *P. aeruginosa*, GCF_000737615.1 for *S. aureus*, GCF_002055965.1 for *Bacillus subtilis*. Read summarization was performed using the featureCounts (2.0.1)[56]. FeatureCounts takes the GTF format annotation file and the alignment file (a.bam file) as input to assign mapped reads to genomic features and then counts reads. The GenBank assembly accession of GTF format annotation files was provided above. The UMI count for each gene was determined using the UMI tools (1.1.2)[57].

### Identification of the putative cells

To determine the number of putative bacteria in each sample, first, the total number of detected genes was determined for each possible barcode. Barcodes are ordered from the largest to the smallest total number of detected genes and numbered in order. We plotted the log10(total number of detected genes per barcode) versus log10(Barcode number) rank plot for each possible barcode. On the rank plot, the knee point (red dashed line) indicated the threshold for putative cells. The knee point was identified according to the

calculation results by STAR solo module in STAR (2.7.10a)[55], which used the EmptyDrop-like type of filtering, and the 10 numeric parameters: nExpectedCells (1500), maxPercentile (0.99), maxMinRatio (10), indMin (45000), indMax (90000), umiMin (300), umiMin-FracMedian (0.01), candMaxN (20000), FDR (0.01), and simN (10000). Barcodes on the left of the threshold were identified as bacteria, but the rest was identified as noise. Only the barcodes identified as bacteria were used for downstream analysis.

## Cell quality control
The total number of UMI counts and detected genes in every single cell were counted. A violin plot combined with a box plot was used to visualize the distribution of UMI count and detected genes of every single cell. Statistical medians of UMI count and detected genes were used to describe the datasets. The most extreme outliers were filtered. A summary of the datasets by smRandom-seq was provided including the cell numbers, thresholds, reads numbers, UMI counts, and detected genes for different biotypes (Supplementary Data 2).

## Analysis of gene expression matrix
The single-bacterium gene expression matrix of filtered data was then analyzed by the Scanpy toolkit (version 1.8.2)[58], including preprocessing, visualization, clustering, and differential expression testing. We kept genes that were expressed in at least three cells and cells that have at least 10 genes detected. We used "pandas.merge" function to create a data frame of different samples, "scanpy.pp.normalize_total" function (target_sum = 1e4) to normalize each cell by total counts over all genes, "scanpy.pp.log1p" function with default parameters to logarithmize the data matrix, "sc.pp.highly_variable_genes" function (min_mean = 0.0125, max_mean = 3, min_disp = 0.5) to identify highly variable genes, "sc.pp.scale" function (max_value = 10) to scale each gene to unit variance and zero mean, "sc.tl.pca" function (svd_solver = 'arpack') to perform principal component analysis (PCA), "scanpy.pl.pca_variance_ratio" function to plot the variance ratio and identify the number of PCs, "sc.pp.neighbors" function for clustering at a resolution of 1, "sc.tl.umap" function to reduce the dimensionality with UMAP and bin the cells into cell populations using Leiden clustering, "sc.tl.rank_genes_groups" function to find marker genes enriched in each cluster with a statistical test (t-test). The "sc.tl.rank_genes_groups" function returned scores underlying the computation of a p-value for each gene for each group. The marker gene was set to a p-value < 0.05 and ordered according to scores. The top 20 marker genes were retained as the DEGs. The expression of the top 5 or 10 DEGs was visualized using a dot plot. The "enrichGO" function of clusterProfiler R package (version 3.16.0) was used for GO enrichment analysis (p-value cutoff value = 0.05 and q-value cutoff value = 0.05) of the top 20 DEGs of each group in R studio (version 4.0.1). The ggpubr R package (version 0.4.0) was used to perform a correlation test and plot the correlation coefficient and the significance level in R studio (version 4.0.1). OriginPro 2021 (9.8.0.200) was used for the data analysis and graphing (Fig. 2d, h, j).

## Saturation analysis
The rRNA-depleted and deeper sequenced E. coli samples were used to generate the saturation curve. The saturation curve plot was generated by randomly selecting the corresponding number of raw reads from the sample library and then using the same alignment pipeline to calculate the median numbers of genes detected per cell. Sequencing reads were randomly subsampled to 22k, 18k, 13k, 9k, 4k, 2k, and 1k reads per cell using the Seqtk tool.

## Gene body coverage analysis
RSeQC86[59] geneBody_coverage.py program was used to generate the gene body coverage plots. The BAM file containing aligned, deduplicated reads and the reference BED file generated by converting the comprehensive gene annotation GFF3 file were input to the RSeQC86.

## Cell viability analysis
At 0, 1, 2, 4 h time points after 12.5 μg/mL AMP treatment, E. coli samples were sampled and added to fresh LB medium supplemented with 0.5 μM of propidium iodide (PI) dye. After a 5-min incubation at 37 °C, cells are washed by PBS for three times, followed by fixation with 4% PFA for 15 min. We visualized and quantified the numbers of PI-positive cells and total cells, respectively. We captured images using Leica DMi8 fluorescent microscope.

## Cell respiratory activity analysis
At 0, 1, 2, 4 h time point after 12.5 μg/mL AMP treatment, E. coli samples were sampled and added with fresh LB medium supplemented with 2.5 mM of 5-cyano-2,3-ditolyltetrazolium chloride (CTC) dye. After a 30-min incubation at 37 °C, cells are washed with PBS three times, followed by fixation with 4% PFA for 15 min. Total CTC fluorescence intensity was detected using Tecan SPARK fluorescence microplate reader (Ex/Em = 480/630 nm). CTC fluorescence intensity was normalized for cell density. We captured images using Leica DMi8 fluorescent microscope. The duplicate numbers of samples are indicated in figure legends. Results of E. coli samples used in cell viability and respiratory activity analysis experiments are presented as means ± standard deviation.

## Statistics and reproducibility
Statistical details for each experiment are provided in the figure legends. Two-tailed Student's t-tests were used for the calculation of p values. Statistical significance was established at a 95% confidence level (p-value < 0.05). The microfluidic encapsulation experiment, beads synthesis experiment, and micrographs experiment were repeated at least four times independently with similar results. Validation experiments of smRandom-seq were repeated three times independently. No statistical method was used to predetermine the sample size. No data were excluded from the analyses. The experiments were not randomized. The Investigators were not blinded to allocation during experiments and outcome assessment.

## Reporting summary
Further information on research design is available in the Nature Portfolio Reporting Summary linked to this article.

## Data availability
The smRandom-seq data generated in this study have been deposited in the Genome Sequence Archive database under accession code CRA011274. The public B. subtilis data used in this study is available in the Gene Expression Omnibus database under accession code GSE217916. The public E. coli data used in this study is available in the Gene Expression Omnibus database under accession code GSE168963. Source data are provided with this paper.

## Code availability
All script files used in the analysis in this manuscript can be downloaded from GitHub at are available at https://github.com/wanglab2023/smRandom-seq[60].

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

## Acknowledgements

The project was supported by the National Natural Science Foundation of China (No. 32200073, Y.W., No. 82200977, Z.X.), Leading Innovative and Entrepreneur Team Introduction Program of Zhejiang (No. 2021R01012, Y.W.), and Liangzhu Laboratory. This work was supported by Harvard MRSEC Program under NSF award DMR 14–20570. This work was also supported by the Wyss Institute for Biologically Inspired Engineering at Harvard University. Thanks for the technical support by the core facilities and computing platform of Zhejiang University Medical Center and Liangzhu Laboratory. We thank the helpful comments from Peter Lu.

## Author contributions

Y.W. and D.W. conceived the study. Y.W., Z.X., K.S., Y.T.W., N.L., and Y.H. developed the chemistry for smRandom-seq. Y.W., Y.L., S.Z., and T.C. constructed the microfluidics platform. Y.T.W., R.R., X.H., M.S., Y.J., and Y.Y. contributed to the antibiotic-treated bacteria experiments. Z.X., T.Z., J.C, Y.S., Y.C., G.G., and Z.W. analyzed the data. Z.X. and Y.W. wrote the manuscript. Y.W., D.W., and P.Y. supervised this project. All authors have revised and approved the final manuscript.

## Competing interests

The authors declare no competing interests.
