## [Peer Review File · Nature Communications]

Reviewers' Comments:

Reviewer #1:

Remarks to the Author:

Xu et al describe a droplet microfluidic platform for single bacteria RNA seq. They implemented a powerful ribosomal RNA depletion method and show a performance (e.g. in terms of number of detected genes per cell) that is beyond previous combinatorial indexing methods. The paper is hence of significant interest for the community and I recommend publication after addressing a few minor points:

Figure 2.: As a first demonstration and illustration of the new method, authors should provide a multidimensional reduction plot such as UMAP for all bacterial species shown in c). Do all these very distinct species cluster separately? This information would be very useful, before analyzing differentially expressed genes upon antibiotics treatment within one and the same species (Fig. 3).

In addition to the GFP expression experiment shown in Supplemental Figure 4, the paper would greatly benefit from further functional validation of the obtained transcriptomic signatures. This could e.g. be done by correlating transcriptomic signatures with measured cell viabilities (e.g. for the sample shown in main text Fig. 4) or, even better, with measured resistances to antibiotics (by e.g. processing sensitive/ resistant cells in both assay types). Would it be possible to use hashtag antibodies for being able to overlay transcriptomic and functional (phenotypic assay) data sets? Even if not, more functional validation should be done.

Discussion: In contrast to Petri-Seq and MicroSPLIT the approach shown here would probably also allow to monitor transcriptomic changes upon exposure to drugs in droplets (shown for e.g. mammalian cells in Mathur L et al., Nature Communications 2022). This could be a very interesting application for functional profiling of drugs on human microbiota and should be discussed here.

How many cells can be processed per experiment? I could not find any clear statement on the throughput of smRandom-seq, but it seems to be significantly lower as compared to Petri-Seq and MicroSPLIT. This should be openly discussed in quantitative terms.

Reviewer #2:

Remarks to the Author:

The authors report a new method (smRandom-seq) for single-cell RNA-seq in bacteria. Their method relies on an optimized microfluidic droplet barcoding platform which allows high-throughput RNA-seq on single-cell level. In addition, the implementation of a Cas9-based ribosomal RNA depletion protocol increases both the throughput and sensitivity of scRandom-seq.

The manuscript by Xu et al. is technically oriented and focuses on the workflow as well as validation of a new droplet barcoding platform for single-cell RNA-seq in bacteria using random primer and poly-A tailing prior to barcoding. To validate smRandom-seq, the authors primarily use *E. coli* and *B. subtilis*. Further proof-of-principle experiments were performed in several relevant Gram-negative and Gram-positive bacteria. *E. coli* was used for all further method development including the integration of the Cas9-based ribosomal RNA depletion step. Using the improved smRandom-seq protocol, the authors further analyzed single-cell gene expression patterns in *E. coli* under antibiotic exposure of ciprofloxacin and ampicillin. This allowed them to identify several subpopulations with distinct expression patterns of SOS and ROS response genes as well as metabolic activity upon antibiotic treatment.

Overall, the method presented by Xu et al. would be of high interest and a valuable addition to the burgeoning field of prokaryotic scRNAs-seq. However, further experiments are required to prove broad applicability and robustness of the workflow. Due to missing information and lack of clarity, it'd be very hard to independently reproduce the results shown. Further, the figures require some rework with a focus on layout and specificity to provide clarity of the individual panels. In general, it is hard to follow the flow of the manuscript as important details are missing in several figure legends and for some experiments the intention of is not clear. As further addressed below,

common nomenclature and scientific style should be applied throughout the entire text, figures, and supplementary material.

Major points:

- One of the advantages mentioned by the authors is the ideal length of the cDNA libraries obtained from smRandom-seq with about 200-500 bp without any fragmentation. This raises the question whether full transcripts can be captured with this method (many bacterial genes are transcribed as polycistrons with the mRNAs then often being thousands of nucleotides in length). The authors should specifically state the actual length of the mRNA-derived insert in their libraries, show coverage plots, and discuss potential biases in their genome coverage.
- Figure 2a shows a doublet rate of only 1.6%. Therefore, multiplets within the same species have not been considered although these are likely to play an important role here. Given that *E. coli* was sampled in excess compared to *B. subtilis*, the probability to generate multiplets from the same species is even higher. Since the authors showed that smRandom-seq can be applied to various species, it would be important to do doublet rate determination in a multi-species context, as the likelihood of multiplets from the same species would be drastically reduced. Further the authors should comment on the shift of number of UMIs from *B. subtilis* compared to *E. coli* and the uneven numbers of cells per species. According to the methods section, equal numbers of bacteria were used as input for smRandom-seq.
- The text related to figure 2c suggests that these experiments did not include any rRNA depletion. As nicely shown in the following panels, smRandom-seq shows high efficiency and RNA capture on a single-cell level in *E. coli*, but this improvement is mainly based on the introduction of the rRNA depletion step. However, the improved method was only performed in *E. coli*, therefore applicability to multiple bacteria species is not proven as claimed by the authors. To show the broad applicability of their method, datasets including different bacterial species and even mixed populations from panel c including rRNA depletion are necessary.
- Successful Cas9-based depletion of rRNA-derived reads from bacterial scRNA-seq libraries was recently demonstrated in a preprint from another group (www.biorxiv.org/content/10.1101/2022.11.28.518171v1), the two manuscripts probably overlapped. Suggest to include a brief comparison of these two methods using Cas9.
- The data shown in Figure 2g reveals that there is a limitation in throughput at around 7600 cells. Here the authors should discuss why a higher number of cells leads to a drop in the number of detected genes. This said, it is surprising that the authors see further potential for increasing the throughput using their method as they've shown here a clear limitation. Due to this limitation and the data shown in panel h it is also not clear how a median of 1045 genes can be obtained in such a high number of cells as shown in panel k. According to panel g the median number of genes should be lower.
- Some data obtained in *E. coli* upon ampicillin treatment don't fit the authors' explanations in the results section. For example, the authors comment that *aceA* shows high expression at the 1h time point (subclusters 1, 3 and 4); yet data shown in Fig. 4g shows the highest upregulation in subcluster 8 which is associated with the 2 h time point. Similar inconsistency exists for panel k (*secY*). *secY* is even higher expressed in subcluster 2 compared to 6.
- The authors advocate smRandom-seq as an "easy-to-use" method. Thus, reproducibility and availability of the equipment and data set should be straightforward. Therefore, we would expect that all required equipment, customized protocols and data analysis pipelines are readily available, and that the platform does not need heavy investment for a lab to implement it. However, details on the availability of the customized droplet barcoding platform as well as the required equipment are missing, and so are details of data and code availability.
- There are only 15 lines of text describing the bioinformatics work in this paper. This text does not cite any tools and does not mention any specific thresholds or cutoffs applied. To fully understand the results, this is required.
- Continuing from the above, the data analysis is lacking important information about UMIs, detected genes and gene count thresholds. There is also no mention about how many cells were sequenced, the filtering steps and how many cells were ultimately removed based on filtering. This is important because they would like to compare their method to other droplet-based protocols and this efficiency is needed to provide a fair comparison.

Minor points:

General

- microSPLIT instead of microSPLIT-seq (throughout the entire text).
- Gram-negative / -positive instead of gram-negative / - positive (throughout the entire text).
- Abbreviations should be explained the first time they're used (e.g. DEGs, UMI, GO).
- Abbreviations used in figures solely need to be explained in figure legends (e.g. TCA).
- Units should be harmonized throughout the entire text (e.g. ug or µg etc.)
- A statement about the data availability (GEO accession) is required.
- With a new method, we'd expect to see a supporting link to GitHub or similar code repository so interested researchers could reproduce their findings
- The authors need to define and clarify the terminology: UMIs, detected genes and gene counts - in its current form they are used interchangeably and not consistently like other single cell papers and methods. It appears the authors looked at all three numbers and where appropriate, picked the highest number and reported this value.

Introduction

- The statement that "scRNA-seq methods have been developed and widely used in eukaryotic biology" should be supported by a suitable reference.
- The statement that "rRNA occupies >80% of total bacterial RNAs" should be supported by a suitable reference.
- Rosenberg et al. 2018 (DOI: 10.1126/science.aam8999) should be cited as the original publication of split-pool barcoding strategy.
- "A few subpopulations" - this is too general, please either refer to the actual number or proportion

Results

- Long version of PFA is missing in the text.
- The list of designed gRNAs should be provided as a supplementary table.
- The source data of Fig. 2d should be provided as a supplementary table as some of the RNA classes are not visible in the shown figure.
- Gene names and species names should be in italics throughout all figures and text.
- "bacteria were imaged to confirm single bacterial morphology and counted" - how were the bacteria counted. Was this manually or through computer software?
- The statement that "fluoroquinolones are potent inducers of the SOS response" should be supported by a reference.
- "The expression levels of metabolic genes were significantly reduced with treatment time,..."
- "...consistent with the reduced detected UMIs and genes..."
- The authors report a homogeneous morphology of *E. coli* even after 4 hours treatment of AMP at 5 µg/ml though Supplementary Fig. 8b shows a clear elongation of several cells after 4 hours of treatment.
- Throughout the results, there are limited significance thresholds or statistical tests to confirm many of their general statements. Also, many of the comments are too general. For example, "detected genes per bacterium were all about 200". "almost doubled". "apparent DEG". "algorithmically clustered".

- Figure 1:

- o Consider to move panels b-e to the supplement as the experimental setup is the most relevant part here.
- o Scaling in panel b and d are not readable.
- o "Break drops" - should this be break droplets?

- Figure 2:

- o Several typos and missing spaces in the figure. E.g., *A. baumannii*; box plot.
 - o Panel b is missing in the figure legend and the y-axis label should be "species specificity".
 - o *B. subtilis* should be included in panel c for comparison.
 - o If mention ncRNA's are detected, but they are not visible in the plot
 - o The axis labels of panel f are incomplete.
 - o E1-E4 in panel j should be explained as well as arrows above and next to the violin plots. E1-E4 should also be explained in the text, accordingly.
- The box plot is missing; instead, medians and standard deviations are shown.
- o Axis labels should be harmonized throughout the entire figure. E.g., gene count or gene counts;

number of UMIs or median UMIs.

o It is not clear how many single-cells of *E. coli* were analyzed for the individual panels (e.g. panel d, e, f...). This should be clarified in the figure as well as in the referring text.

o 2k - it seems strange due to the heterogeneous nature of cells that they cluster like this - it's almost too perfect. We'd expect to at least see some sub clusters evident. How was the data transformed prior to this - this would greatly help.

- Figure 3:

o Panel d is missing a y-axis label (expression levels).

o In panel f it is not clear what the gene ratio exactly means and what counts are. Further details in the figure legend would be helpful. Besides, top three counts are cropped.

o It is not clear to what the RNA-seq data shown in panel e, g, h was normalized to.

- Figure 4:

o Cluster numbers should be more clearly highlighted and adjusted properly in panel c. Especially subclusters 2 and 4. Further, it is not clear what "leiden" means above.

o Label of x-axis is missing in panel d.

o The color code for the mean expression is missing in panel e and the data obtained in this panel should be better described in the results section.

o The scaling of figures f-m should be adjusted to allow direct comparison. Further the information about normalization is missing and it is not clear why the expression levels are even negative in some panels.

o The labeling is off for panel k-m. Panel m is supposed to be a t-SNE plot according to the figure legend and not a UMAP for the cell division marker gene *ftsN*.

o Dashed red lines used in panel l-o should be explained in the figure legend. Subcluster 6 should not be highlighted in the panel or according to the results section.

Discussion

- "...which are usually not of interest."

- The mentioned issue with the template-switching strategy should be supported by suitable references.

- The authors comment on the integration of a pre-indexing step. This should be explained in more details how this could be technically solved and integrated. Addressed advantages would be the option to process multiple samples and the reduction of batch-effects. Though the data in figure 2 showed a clear limitation in sample throughput but at the same time no batch effects at all.

Material and Methods

- Details about the used reverse transcriptase kit should be given.

- Further details should be given for the barcode beads synthesis, especially as this was a customized and adapted step compared to previous work.

- The information about the used ratio of the cDNA : Cas9 enzyme : sgRNA should be added.

- Details about the used reference genomes and annotation files for the data analysis are missing.

- "annotations were performed using featureCounts" - I'm not sure featureCounts annotates. It counts reads from an alignment file such as a .bam or .sam file.

- You mention the tool Origin that was used for data analysis and graphing - there is no reference so we can't determine the specific tool - we assume this was just used with graphing. Can you comment on why this tool was used instead of R, which was also clearly used.

Supplementary Material

- Supplementary Fig. 3: We assume that the data shown in panel a and b were obtained without rRNA depletion. However, it is not fully clear and should be clarified in the figure legend. Further, the authors should comment why they detected much higher numbers of genes in *B. subtilis* compared to *E. coli* as well as to all other tested species in Fig. 2c.

- Axis labels in Supplementary Fig. 5b are incomplete.

- Panel a can be skipped in Supplementary Fig. 6 as there is only a single panel. Same applies for the text. Consider changing the x-axis label to minutes or hours.

- Supplementary Fig. 8 b includes still underline from spell check.

- It is not clear where in the text/figures some of the supplementary tables are linked to. The

supplementary tables mentioned in the text should also be labeled accordingly.

Reviewer #3:

Remarks to the Author:

This paper reports a droplet-based method for scRNA-seq of bacteria. The results look promising but not enough information is provided to put them into context.

1. Several important references are missing from the manuscript. First, there already are high throughput methods for scRNA-seq of bacteria that use some form of droplet microfluidics. Second, there are at least two reports that apply rRNA depletion in the context of bacterial scRNA-seq. These papers need to be referenced and appropriately compared to:

Droplet based scRNA-seq for bacteria:

- McNulty et al. Droplet-based single cell RNA sequencing of bacteria identifies known and previously unseen cellular states
<https://www.biorxiv.org/content/10.1101/2021.03.10.434868v2.abstract>
- Wang et al. "Massively-parallel Microbial mRNA Sequencing (M3-Seq) reveals heterogeneous behaviors in bacteria at single-cell resolution"
<https://www.biorxiv.org/content/10.1101/2022.09.21.508688v2>
- Ma et al. "Bacterial droplet-based single-cell RNA-seq reveals heterogeneity in bacterial populations and in response to antibiotic perturbation"
<https://www.biorxiv.org/content/10.1101/2022.08.01.502326v1.abstract>
-

There are two papers that combine rRNA depletion with bacterial scRNA-seq:

- Homberger et al. "Improved bacterial single-cell RNA-seq through automated MATQ-seq and Cas9-based removal of rRNA reads"
<https://www.biorxiv.org/content/10.1101/2022.11.28.518171v1.full.pdf>
- Wang et al. "Massively-parallel Microbial mRNA Sequencing (M3-Seq) reveals heterogeneous behaviors in bacteria at single-cell resolution"
<https://www.biorxiv.org/content/10.1101/2022.09.21.508688v2>

2. The data analysis section needs to be expanded and analysis code should be provided on GitHub or similar. For example, authors need to include information about which genomes were used for alignment, how multiply-aligning reads were handled, how down-sampling analysis was performed etc.

3. Minor: PETRI-seq and microSPLiT-seq also use random reverse transcription primers and perform RT inside fixed cells.

4. My biggest concern is around the reporting of gene and UMI counts. There are several places in the paper where the authors claim that they detect on the order of 1000 genes per cell. (E.g. "A median of ~1000 genes can be detected per bacterium in a population of ~8000 E. coli.") However, it is not entirely clear whether this is an actual measurement or a computational extrapolation. If the authors indeed (experimentally) detect 1000 genes/cell that would be a great improvement over the state of the art. However, the experimental data in Fig. 2C shows a median gene count of 225 which is much more modest but would be in line with gene counts reported in earlier bacterial scRNA seq papers. Please make it clear which numbers are measurements and which are extrapolation.

5. Fig. 2i lists 1045 genes detected/cell for E. coli at ~50K reads but Fig. 2h shows 1045 genes detected at ~22K reads.

6. Fig. 2h caption says "Number of genes detected per E. coli when down sampling total read counts to the indicated depths." Please provide a description of how this down sampling analysis is performed and which experimental data were used.

7. For a different experiment the paper reports "We detected a median count of 6564 and 1785 UMIs per cell for *B. subtilis* and *E. coli*, as well as a median count of 1249 and 429 genes per cell, respectively (Supplementary Fig. 3a, b)." Please report rRNA and mRNA UMIs separately. Also, given that the *E. coli* gene count is about twice that reported in Fig. 2C, is this a result of differences in sequencing depth? How many cells were used in that experiment?
8. Are the data in Fig. 2e from a different experiment than those in 2C? The UMI counts (before depletion) are different.
9. Fig. 2e and Supplementary fig. 3c: Does each dot correspond to a barcode is a barcode a putative cell?
10. Please provide a summary of the experiments performed including the number of cells, the number of raw sequencing reads per cell, the number or mRNA UMIs, rRNA UMIs, genes etc.
11. Minor: What is the purpose of comparing cell numbers in Fig. 2i? Both PETRI-seq and microSPLIT were used to perform experiments with many more cells then listed.
12. Fig. 1a implies that multiple UMIs could be derived from the same transcript.

Replies to Reviewers' Comments:

Reviewer #1:

Xu et al describe a droplet microfluidic platform for single bacteria RNA seq. They implemented a powerful ribosomal RNA depletion method and show a performance (e.g. in terms of number of detected genes per cell) that is beyond previous combinatorial indexing methods. The paper is hence of significant interest for the community and I recommend publication after addressing a few minor points:

Reply: We are grateful to reviewer's effort to review our paper. We have fully addressed the concerns as follows.

1. Figure 2.: As a first demonstration and illustration of the new method, authors should provide a multidimensional reduction plot such as UMAP for all bacterial species shown in c). Do all these very distinct species cluster separately? This information would be very useful, before analyzing differentially expressed genes upon antibiotics treatment within one and the same species (Fig. 3).

Reply: We would like to thank the reviewer's kind suggestion. We provided a UMAP plot (Supplementary Fig. 5b) for all bacterial species shown in Figure 2c. The UMAP plot showed that all these very distinct species cluster separately. However, the single-species experiments of these bacterial species in Figure 2c were performed separately. We added a three-specie mixture experiment of *A. baumannii*, *K. pneumoniae*, and *E. coli* (Supplementary Fig. 4a), and provide a UMAP plot for this three-specie mixture data (Supplementary Fig. 4b). The UMAP plot also showed that all these species in the three-specie mixture clustered separately (Supplementary Fig. 4b).

Supplementary Figure 5

Supplementary Figure 5b, UMAP plot colored by species, *E. coli*, *A. baumannii*, *K. pneumoniae*, *P. aeruginos*, and *S. aureus*, identified from single-species experiments.

Supplementary Figure 4

Supplementary Figure 4. Three-species-mixing experiment. a, Experimental design for three-species mixture including *A. baumannii*, *K. pneumoniae*, and *E. coli* by smRandom-seq. b, UMAP plot colored by species, *A. baumannii*, *K. pneumoniae*, *E. coli* and mixed cells, identified from the three-species mixture.

2. In addition to the GFP expression experiment shown in Supplemental Figure 4, the paper would greatly benefit from further functional validation of the obtained transcriptomic signatures. This could e.g. be done by correlating transcriptomic signatures with measured cell viabilities (e.g. for the sample shown in main text Fig. 4) or, even better, with measured resistances to antibiotics (by e.g. processing sensitive/ resistant cells in both assay types). Would it be possible to use hashtag antibodies for being able to overlay transcriptomic and functional (phenotypic assay) data sets? Even if not, more functional validation should be done.

Reply: We thank the reviewer for the valuable comments and suggestions. We measured the cell viabilities of *E. coli* upon ampicillin (AMP) treatment using the dye propidium iodide (PI), which selectively stains dead cells. The results showed that the proportions of dead cells were increased (Supplementary Fig. 15a) and the cell viabilities were decreased after AMP treatment (Supplementary Fig. 15b), consistent with the upregulation of DNA damage related genes (*recA* and *sulA*) in the top 10 differentially expressed genes (DEGs) among groups (Supplementary Fig. 14a). We also used the dye 5-cyano-2,3-ditolyl tetrazolium chloride (CTC) to assess the total and single-cell respiratory activity of *E. coli* upon antibiotics (AMP) treatment. We found that the average respiratory activity of *E. coli* was increased after 1 hour of antibiotic treatment, and then decreased at 2- and 4- hour time points (Supplementary Fig. 16a), which is consistent with the trend of the expressions of metabolic genes first increasing and then decreasing after antibiotics treatment (Supplementary Fig. 14h). A sub-population of bacteria at 4-hour time point showed high levels of CTC fluorescence intensity (Supplementary Fig. 16b), which is consistent with the different expression patterns of metabolic genes in subclusters 6, 9, and 10 of 4-hour time point.

Supplementary Figure 15

Supplementary Figure 15. Cell viability analysis of *E. coli* upon 12.5 µg/mL AMP.

a, Cell viability analysis by propidium iodide (PI). Representative fluorescence and bright filed images of *E. coli* samples at 0h, 1h, 2h, 4h time point after 12.5 µg/mL AMP treatment. Red color (PI) showed positive staining of dead cells. Scale bar showed 10 µm. **b**, Graphic representation of statistical result of cell viability. n=6.

Supplementary Figure 16

Supplementary Figure 16. Cell respiratory activity analysis of *E. coli* upon 12.5 µg/mL AMP.

a, Cell respiratory activity analysis by the 5-cyano-2,3-ditolyltetrazolium chloride (CTC) dye staining in combination with fluorescence microplate reader. Graphic representation of statistical

result of normalized CTC fluorescence intensity (CTC fluorescence intensity/Cell count) of *E. coli* samples at 0h, 1h, 2h, 4h time point after 12.5 µg/mL AMP treatment. n=6. ns: not significant. **b**, Representative fluorescence and bright field images of *E. coli* samples at 0h, 1h, 2h, 4h time point after 12.5 µg/mL AMP treatment. Red color (CTC) showed cells with high cell respiratory activity. Scale bar showed 10 µm.

3. Discussion: In contrast to Petri-Seq and MicroSPLIT the approach shown here would probably also allow to monitor transcriptomic changes upon exposure to drugs in droplets (shown for e.g. mammalian cells in Mathur L et al., Nature Communications 2022). This could be a very interesting application for functional profiling of drugs on human microbiota and should be discussed here.

Reply: We would like to thank the reviewer's good suggestion. We added the discussion of the application of smRandom-seq for functional profiling of drugs on human microbiota in the revised manuscript (**Page 14**): "Moreover, combined with the Combi-Seq¹ that using combinatorial DNA barcoding approach to encode treatment conditions in droplet, smRandom-seq would probably allow monitoring the bacterial transcriptomic changes at single bacterium level for high-throughput drug screening".

4. How many cells can be processed per experiment? I could not find any clear statement on the throughput of smRandom-seq, but it seems to be significantly lower as compared to Petri-Seq and MicroSPLIT. This should be openly discussed in quantitative terms.

Reply: Thanks for the reviewer's careful reading. The droplet microfluidic platform used in smRandom-seq can capture cells at a rate of ~160,000 cells/hour. The throughput of the current smRandom-seq is limited by the diversity of barcoded beads. Our study describes a three-step synthesis method of a library of 442,368 (96×96×48) unique barcodes (**Supplementary Fig. 2c**). The throughput of current smRandom-seq is estimated from the Poisson distribution as described by Zilionis *et. al*². About 10,000 cells can be processed per experiment using current smRandom-seq. The throughput of current smRandom-seq is comparable with the throughput of these two split-pool barcoding based methods, Petri-Seq³ (~10,000 cells) and MicroSPLIT⁴ (~10,000 cells). However, the diversity of barcoded beads in smRandom-seq can be expanded in a straightforward manner by increasing the number of barcode fragments, e.g. 384×384×384, or incorporating an additional primer ligation step, e.g. 96×96×96×96. We added a clear statement on the throughput of smRandom-seq in the revised manuscript (**Page 5, 6**): "The diversity of barcode beads in this study was 442,368 (96×96×48). The throughput of current smRandom-seq is estimated from the Poisson distribution described by Zilionis *et. al*². About 10,000 cells can be processed per experiment using the current smRandom-seq".

Supplementary Figure 2

Supplementary Figure 2c, Barcode beads fabrication procedures.

Reviewer #2:

The authors report a new method (smRandom-seq) for single-cell RNA-seq in bacteria. Their method relies on an optimized microfluidic droplet barcoding platform which allows high-throughput RNA-seq on single-cell level. In addition, the implementation of a Cas9-based ribosomal RNA depletion protocol increases both the throughput and sensitivity of scRandom-seq.

The manuscript by Xu et al. is technically oriented and focuses on the workflow as well as validation of a new droplet barcoding platform for single-cell RNA-seq in bacteria using random primer and poly-A tailing prior to barcoding. To validate smRandom-seq, the authors primarily use E. coli and B. subtilis. Further proof-of-principle experiments were performed in several relevant Gram-negative and Gram-positive bacteria. E. coli was used for all further method development including the integration of the Cas9-based ribosomal RNA depletion step. Using the improved smRandom-seq protocol, the authors further analyzed single-cell gene expression patterns in E. coli under antibiotic exposure of ciprofloxacin and ampicillin. This allowed them to identify several subpopulations with distinct expression patterns of SOS and ROS response genes as well as metabolic activity upon antibiotic treatment.

Overall, the method presented by Xu et al. would be of high interest and a valuable addition to the burgeoning field of prokaryotic scRNAs-seq. However, further experiments are required to prove broad applicability and robustness of the workflow. Due to missing information and lack of clarity, it'd be very hard to independently reproduce the results shown. Further, the figures require some rework with a focus on layout and specificity to provide clarity of the individual panels. In general, it is hard to follow the flow of the manuscript as important details are missing in several figure legends and for some experiments the intention of is not clear. As further addressed below, common nomenclature and scientific style should be applied throughout the entire text, figures, and supplementary material.

Reply: We are grateful to reviewer's effort to review our paper. We have fully addressed the concerns as follows. We provided further experiments to prove broad applicability and robustness of smRandom-seq, added more details for experiments and data processing and revised the nomenclature and scientific style.

Major points:

1. One of the advantages mentioned by the authors is the ideal length of the cDNA libraries obtained from smRandom-seq with about 200-500 bp without any fragmentation. This raises the question whether full transcripts can be captured with this method (many bacterial genes are transcribed as polycistrons with the mRNAs then often being thousands of nucleotides in length). The authors should specifically state the actual length of the mRNA-derived insert in their libraries, show coverage plots, and discuss potential biases in their genome coverage.

Reply: We would like to thank the reviewer's kind suggestion. The length of the cDNA libraries

obtained from smRandom-seq is about 200-500 bp. Cut the total length of the adaptors (123 nt), the actual length of the mRNA-derived insert in libraries of smRandom-seq is about 100~400 bp. We added a statement on the actual length of the mRNA-derived insert in libraries of smRandom-seq in the revised manuscript (Page 6): “After cutting the total length of the adaptors (123 nt), the actual length of the mRNA-derived insert in libraries of smRandom-seq is about 100~400 bp.” The cDNA libraries obtained from smRandom-seq were sufficient for gene expression analysis. However, full-length transcripts of bacterial genes cannot be captured with current smRandom-seq. We provided the genome coverage percent plot of *E. coli* dataset obtained from smRandom-seq (Supplementary Fig. 7a). The median genome coverage percent of *E. coli* dataset obtained from smRandom-seq was 0.093. We assessed potential biases in our genome coverage by determining the average coverage across all transcripts at each percentile of length from 5' to 3' end of the known transcripts based on mapping of transcriptome reads to *E. coli* genome (Supplementary Fig. 7b). The coverage gradually decreases from 5' to 3' end, which can be explained by the 3' to 5' direction of reverse transcription.

Supplementary Figure 7

Supplementary Figure 7. Coverage of smRandom-seq on the deeper *E. coli* samples.

a, Distribution histogram of genome coverage percent per cell of the deeper *E. coli* sample with rRNA depletion. The median of the genome coverage percent per cell was 0.093. **b**, Average percentile coverage across all transcripts (5' to 3').

2. Figure 2a shows a doublet rate of only 1.6%. Therefore, multiplets within the same species have not been considered although these are likely to play an important role here. Given that *E. coli* was sampled in excess compared to *B. subtilis*, the probability to generate multiplets from the same species is even higher. Since the authors showed that smRandom-seq can be applied to various species, it would be important to do doublet rate determination in a multi-species context, as the likelihood of multiplets from the same species would be drastically reduced. Further the authors should comment on the shift of number of UMIs from *B. subtilis* compared to *E. coli* and the uneven numbers of cells per species. According to the methods section, equal numbers of bacteria were used as input for smRandom-seq.

Reply: Thanks for the reviewer’s careful reading. We agree with that the doublet rate of 1.6% is the inter-species doublet rate. The inter-species doublet rate is widely used to evaluate the crosstalk of other single cell RNA-seq methods, e.g., inDrop⁵, Drop-seq⁶, and 10X Genomics⁷. To compare the performance of smRandom-seq with other methods, we showed the inter-species doublet rate (1.6%). We revised the description of inter-species doublet rate (1.6%) in the manuscript (**Page 6**). It is difficult to distinguish the multiplets within the same species and calculated the total doublet rate. However, it can be statistically inferred that the total doublet rate is about twice the inter-species doublet rate. We added a three-species-mixing experiment of a 1: 1: 1 mixture of *A. baumannii*, *K. pneumoniae*, and *E. coli* (**Supplementary Fig. 4a**). The doublet rate determination in three-species mixture context was about 2.8 % (**Supplementary Fig. 4b**). We also agree with that there was bias in species-mixing experiment by smRandom-seq due to the variations in bacterial size, RNA content and cell wall composition. We are working on reducing the bias in smRandom-seq so that it can be used in microbiome. We added the comment on the bias in the revised manuscript (**Page 6**): “However, the current smRandom-seq used in species-mixing experiments showed bias in cell number, UMI count, and gene count due to variations in bacterial size, RNA content, and cell wall composition.”

Supplementary Figure 4

Supplementary Figure 4. Three-species-mixing experiment. a, Experimental design for three-species mixture including *A. baumannii*, *K. pneumoniae*, and *E. coli* by smRandom-seq. b, UMAP plot colored by species, *A. baumannii*, *K. pneumoniae*, *E. coli* and mixed cells, identified from the three-species mixture.

3. The text related to figure 2c suggests that these experiments did not include any rRNA depletion. As nicely shown in the following panels, smRandom-seq shows high efficiency and RNA capture on a single-cell level in *E. coli*, but this improvement is mainly based on the introduction of the rRNA depletion step. However, the improved method was only performed in *E. coli*, therefore applicability to multiple bacterial species is not proven as claimed by the authors. To show the broad applicability of their method, datasets including different bacterial species and even mixed populations from panel c including rRNA depletion are necessary.

Reply: We would like to thank the reviewer's good suggestion. We performed the Cas9-based rRNA depletion in other bacterial species, including *A. baumannii*, *K. pneumoniae*, *E. coli*, and *S. aureus*, and a mixed population of *A. baumannii*, *K. pneumoniae*, and *E. coli*. The rRNA-depletion treatment reduced the rRNA proportion from 71% to 10%, 88% to 29%, 74% to 4%, 91% to 45%, 83% to 5%, respectively (Supplementary Fig. 6b-f), which proved the broad applicability of our method.

Supplementary Figure 6

Supplementary Figure 6b-f, Representative proportions of transcript categories of single bacterial species, including *A. baumannii* (b), *K. pneumoniae* (c), *P. aeruginosa* (d), and *S. aureus* (e), and a mixture of *A. baumannii*, *K. pneumoniae*, and *E. coli* (f) with (Del-rRNA) and without rRNA depletion (CON). other, all other RNA classes.

4. Successful Cas9-based depletion of rRNA-derived reads from bacterial scRNA-seq libraries was recently demonstrated in a preprint from another group (www.biorxiv.org/content/10.1101/2022.11.28.518171v1), the two manuscripts probably overlapped. Suggest to include a brief comparison of these two methods using Cas9.

Reply: We would like to thank the reviewer's good suggestion. Both these two methods use a Cas9-based strategy (DASH) to deplete rRNA-derived reads from bacterial scRNA-seq libraries. Homberger *et al.* applied this rRNA depletion strategy on a low-throughput bacterial scRNA-seq method. However, we applied it on the high-throughput bacterial scRNA-seq method, smRandom-

seq. We added a brief comparison of this method and our smRandom-seq in the revised manuscript (Page 13): “Recently, Homberger *et al.* also posted successful Cas9-based rRNA depletion for the libraries of *Salmonella* by a low-throughput bacterial single cell RNA-seq method¹⁸.”

5. The data shown in Figure 2g reveals that there is a limitation in throughput at around 7600 cells. Here the authors should discuss why a higher number of cells leads to a drop in the number of detected genes. This said, it is surprising that the authors see further potential for increasing the throughput using their method as they’ve shown here a clear limitation. Due to this limitation and the data shown in panel h it is also not clear how a median of 1045 genes can be obtained in such a high number of cells as shown in panel k. According to panel g the median number of genes should be lower.

Reply: Thanks for the reviewer’s careful reading. The **Figure 2g** is a Gene count vs Barcode plot, which is widely used in the pre-processing of scRNA-seq data to identify putative cells from noise. In droplet-based single cell RNA-seq platform, there will be noise in the library due to the bacteria-empty droplets, wrong barcodes on beads and etc. The Gene count vs Barcode plot displays the gene count of each barcode that was rank sorted by gene count. The sharp drop of gene counts indicates a clear separation of cellular barcodes from noise. The barcodes with low gene counts were classified as noise. We loaded ~10,000 cells in the experiment of **Figure 2g** and identified 7645 putative bacterial barcodes. About 10,000 cells can be processed per experiment using current smRandom-seq. The throughput of the microfluidics-based smRandom-seq depend on the diversity of barcode beads. The diversity of barcode beads in this study was 442,368 (96×96×48). The diversity of barcode beads can be expanded by increasing the number of barcode fragments, e.g. 384×384×384, or incorporating an additional primer ligation step, e.g. 96×96×96×96. As mentioned above, the Gene count vs Barcode plot in **panel g** was only used to identify putative cells from noise in our previous analysis. For the saturation analysis in **panel k**, we increased sequencing data volume of the rRNA-depleted *E. coli* library in **panel g**. A median of 1045 genes in 7645 cells were detected in the larger data with a sequencing depth of 22k mapped reads per cell. To avoid confusion, we replotted **panel g** with the larger data (**Figure 2g**).

Figure 2g, Gene count versus barcode rank plot. Identification of the barcodes that represent actual bacteria. The *E. coli* sample with rRNA depletion was sequenced more deeply for the next performance evaluations. Barcodes of the deeper *E. coli* sample with rRNA depletion are ordered

from the largest to smallest gene count. On the gene count versus barcode rank plot, the knee point (red dashed line) indicates the threshold for actual *E. coli* sample (Cell number: 7645).

6. Some data obtained in *E. coli* upon ampicillin treatment don't fit the authors' explanations in the results section. For example, the authors comment that *aceA* shows high expression at the 1h time point (subclusters 1, 3 and 4); yet data shown in Fig. 4g shows the highest upregulation in subcluster 8 which is associated with the 2 h time point. Similar inconsistency exists for panel k (*secY*). *secY* is even higher expressed in subcluster 2 compared to 6.

Reply: Thanks for the reviewer's careful reading. We agree with that the expression patterns of *aceA* or *secY* in *E. coli* upon ampicillin treatment were not accurately described. We previously commented that *aceA* showed high expression at the 1h time point (subclusters 1, 3 and 4) and *secY* showed high expression at in subcluster 2 according to the mean expression levels of these genes in the dot plot (Fig. 4e). We revised the description of the expression levels of these two genes in subclusters according to the feature expression of these genes in each bacterium in low-dimensional space (UMAP) (Fig. 4g, k) in the manuscript (Page 10). *aceA* was a DEG of subcluster 1 and 4 of 1h time point (Fig. 4e) and showed general high expression in 1h time point (Fig. 4g). As time went on, *aceA* showed distinct expression patterns among subclusters, especially the highest upregulation in subcluster 8 of 2h time point (Fig. 4g). *secY* was a DEG of subcluster 9 of 4h time point (Fig. 4e) and showed high expression level in subcluster 9 (Fig. 4k). *secY*, also had high expression level in subcluster 2 of 2-hour time point, suggesting that subcluster 9 might be developed from subcluster 2 (Fig. 4k).

Figure 4

Figure 4e, Top 5 DEGs among subclusters. **g, k**, Expression of glyoxylates shunt related marker gene of subclusters 1, 3 and 4 (*aceA*) (**g**), membrane related marker genes of subclusters 6 and 9 (*secY*) (**k**) overlaid on the UMAP plot.

7. The authors advocate smRandom-seq as an "easy-to-use" method. Thus, reproducibility and availability of the equipment and data set should be straightforward. Therefore, we would expect

that all required equipment, customized protocols and data analysis pipelines are readily available, and that the platform does not need heavy investment for a lab to implement it. However, details on the availability of the customized droplet barcoding platform as well as the required equipment are missing, and so are details of data and code availability.

Reply: Thanks for the reviewer's careful reading. We previously advocated the microfluidic-based smRandom-seq as an "easy-to-use" method because the hardware of smRandom-seq is similar with the widely used 10X Genomics single cell platform. To avoid confusion, we deleted this "easy-to-use" advocacy in the revised manuscript. We added the details on the required equipment, customized protocols, data analysis pipelines, the availability of the customized droplet barcoding platform, data and code availability in the revised manuscript (**Page 17, 18, 22**).

8. There are only 15 lines of text describing the bioinformatics work in this paper. This text does not cite any tools and does not mention any specific thresholds or cutoffs applied. To fully understand the results, this is required.

Reply: We would like to thank the reviewer's good suggestion. We added the description of bioinformatics work in this study (**Page 20, 21**) and the citation of tools, including Cutadapt (version 3.7)⁹, STAR (2.7.10a)¹⁰, featureCounts (2.0.1)¹¹, UMI-tools (1.1.2)¹², and Scanpy toolkit (version 1.8.2)¹³. We also provide a summary (**Supplementary table 4**) including the specific thresholds and cutoffs.

9. Continuing from the above, the data analysis is lacking important information about UMIs, detected genes and gene count thresholds. There is also no mention about how many cells were sequenced, the filtering steps and how many cells were ultimately removed based on filtering. This is important because they would like to compare their method to other droplet-based protocols and this efficiency is needed to provide a fair comparison.

Reply: We would like to thank the reviewer's good suggestion. UMIs represent the molecular barcodes for unique cDNAs. The UMIs for each gene were determined using UMI-tools (1.1.2)¹². Detected genes represent the total number of genes detected in each cell. We identified the gene count threshold according to calculation results by STAR solo module in STAR (2.7.10a)¹⁰. We added the information for data analysis including processing of the sequencing data, identification of the barcodes, and cell quality control in the revised manuscript (**Page 20, 21**). We also provided a summary (**Supplementary table 4**) of the datasets by smRandom-seq including the number of identified cells, the threshold for cell quality control, the number of filtered cells, the median UMIs per cell, and the median genes per cell, etc.

Minor points:

1. General

1.1 microSPLiT instead of microSPLiT-seq (throughout the entire text).

Reply: Thanks for the reviewer's careful reading. We corrected the mistakes in the revised

manuscript (**Page 3, 4, 7, 12, 32**).

1.2 Gram-negative / -positive instead of gram-negative / - positive (throughout the entire text).

Reply: Thanks for the reviewer's careful reading. We corrected the mistakes in the revised manuscript (**Page 4, 6, 15**).

1.3 Abbreviations should be explained the first time they're used (e.g. DEGs, UMI, GO).

Reply: Thanks for the reviewer's careful reading. We explained the abbreviations, including complementary DNA (cDNA), paraformaldehyde (PFA), unique molecular identifier (UMI), differentially expressed genes (DEGs), Gene Ontology (GO), and next generation sequencing (NGS) in the revised manuscript (**Page 4, 5, 9, 11, 13**).

1.4 Abbreviations used in figures solely need to be explained in figure legends (e.g. TCA).

Reply: Thanks for the reviewer's careful reading. We explained the abbreviations used in figures in the revised manuscript (**Page 35**): "TCA: tricarboxylic acid cycle."

1.5 Units should be harmonized throughout the entire text (e.g. ug or µg etc.)

Reply: Thanks for the reviewer's careful reading. We harmonized units in the revised manuscript.

1.6 A statement about the data availability (GEO accession) is required.

Reply: We would like to thank the reviewer's kind suggestion. The smRandom-seq datasets in this study have been deposited in the Gene Expression Omnibus. We will provide the GEO accession as soon as approved.

1.7 With a new method, we'd expect to see a supporting link to GitHub or similar code repository so interested researchers could reproduce their findings

Reply: We would like to thank the reviewer's kind suggestion. We added a supporting link to GitHub in the revised manuscript. All script files used in the analysis in this manuscript can be downloaded from GitHub at are available at <https://github.com/wanglab2023/Msc-RNA-seq>.

1.8 The authors need to define and clarify the terminology: UMIs, detected genes and gene counts - in its current form they are used interchangeably and not consistently like other single cell papers and methods. It appears the authors looked at all three numbers and where appropriate, picked the highest number and reported this value.

Reply: Thanks for the reviewer's careful reading. UMI is an acronym for Unique Molecular Identifier and has been widely used for quantitative RNA copy numbers (e.g., 10X Genomics⁷). Both detected genes and gene counts represent the total numbers of detected genes in every single cell. To avoid confusion, we harmonized the terminology using "gene count" in the revised manuscript. We counted both UMI counts and gene counts in every single cell and provided a

summary of datasets (**Supplementary table 4**). We used UMI count for indicating the specificity (**Fig. 2a, b**) and expression levels (**Fig. 2j**). We used gene count for indicating the sequencing performance (**Fig. 2c, e, g, h, i**).

2. Introduction

2.1 *The statement that “scRNA-seq methods have been developed and widely used in eukaryotic biology” should be supported by a suitable reference.*

Reply: Thanks for the reviewer’s careful reading. We added several references^{7, 14-17} for this statement in the revised manuscript (**Page 3**).

2.2 *The statement that “rRNA occupies >80% of total bacterial RNAs” should be supported by a suitable reference.*

Reply: Thanks for the reviewer’s careful reading. We added a reference¹⁸ for this statement in the revised manuscript (**Page 3**).

2.3 *Rosenberg et al. 2018 (DOI: 10.1126/science.aam8999) should be cited as the original publication of split-pool barcoding strategy.*

Reply: Thanks for the reviewer’s careful reading. We added this reference¹⁹ as the original publication of split-pool barcoding strategy in the revised manuscript (**Page 3**).

2.4 *“A few subpopulations” - this is too general, please either refer to the actual number or proportion*

Reply: We would like to thank the reviewer’s kind suggestion. We replaced “A few subpopulations” with actual number (**Three subpopulations**) in the revised manuscript (**Page 3**).

3. Results

3.1 *Long version of PFA is missing in the text.*

Reply: Thanks for the reviewer’s careful reading. We added the long version of PFA (**paraformaldehyde**) in the revised manuscript (**Page 5**).

3.2 *The list of designed gRNAs should be provided as a supplementary table.*

Reply: We thank for the reviewer's suggestion and added the list of designed gRNAs (**Supplementary table 3**).

3.3 *The source data of Fig. 2d should be provided as a supplementary table as some of the RNA classes are not visible in the shown figure.*

Reply: Reply: We thank for the reviewer's suggestion and added the source data of **Fig. 2d (Source Data Fig 2d)**.

3.4 *Gene names and species names should be in italics throughout all figures and text.*

Reply: Thanks for the reviewer's careful reading. We checked the gene names and species names in all figures and text.

3.5 *"bacteria were imaged to confirm single bacterial morphology and counted" - how were the bacteria counted. Was this manually or through computer software?*

Reply: Thanks for the reviewer's careful reading. Bacteria counts were done manually under a microscope.

3.6 *The statement that "fluoroquinolones are potent inducers of the SOS response" should be supported by a reference.*

Reply: Thanks for the reviewer's careful reading. We added a reference²⁰ for the statement "fluoroquinolones are potent inducers of the SOS response" (**Page 9**).

3.7 *"The expression levels of metabolic genes were significantly reduced with treatment time,..."*

Reply: Thanks for the reviewer's careful reading. We corrected the mistake in the revised manuscript (**Page 9**).

3.8 *"...consistent with the reduced detected UMIs and genes..."*

Reply: Thanks for the reviewer's careful reading. We corrected the mistake in the revised manuscript (**Page 9**).

3.9 *The authors report a homogeneous morphology of E. coli even after 4 hours treatment of AMP at 5 µg/ml though Supplementary Fig. 8b shows a clear elongation of several cells after 4 hours of treatment.*

Reply: We thank the reviewer for the valuable comment. We agree with the reviewer and revised the description (**Page 9**): "All the *E. coli* showed a clear elongation after a 4-hour treatment of AMP at 5 µg/ml (**Supplementary Fig. 8b**)."

3.10 *Throughout the results, there are limited significance thresholds or statistical tests to confirm many of their general statements. Also, many of the comments are too general. For example, "detected genes per bacterium were all about 200". "almost doubled". "apparent DEG". "algorithmically clustered".*

Reply: We thank the reviewer for the valuable comment. We revised these general statements (**Page 6, 7, 10**).

3. Figures

3.1 *Figure 1:*

o Consider to move panels b-e to the supplement as the experimental setup is the most relevant part

here.

Reply: We would like to thank the reviewer's kind suggestion. We moved panels b-e to the supplement (**Supplementary Fig. 1**).

Supplementary Figure 1

Supplementary Figure 1. The microfluidic platform of smRandom-seq.

a, Image of fixed *E. coli* after reverse transcription and dA tailing. Scale bar: 5 µm. **b**, Design of the device for cell, bead and mix reagents encapsulation. Height: 50 µm. **c**, Image of microfluidic barcoding device. Scale bar: 100 µm. **d**, Image of encapsulated droplets. Scale bar: 100 µm. **e**, Electropherogram of amplicon from barcoded single bacterium cDNAs. Lower marker: 20 bp; upper marker: 1k bp.

o Scaling in panel b and d are not readable.

Reply: Thanks for the reviewer's careful reading. We revised the scaling in panel b and d.

o "Break drops" - should this be break droplets?

Reply: Thanks for the reviewer's careful reading. We corrected it in revised Figure 1.

3.2 Figure 2:

o Several typos and missing spaces in the figure. E.g., *A. baumannii*; box plot.

Reply: Thanks for the reviewer's careful reading. We corrected them in revised Figure 2.

o Panel b is missing in the figure legend and the y-axis label should be "species specificity".

Reply: Thanks for the reviewer's careful reading. We added the figure legend of **panel b** ("Species specificity of UMIs in *E. coli* and *B. subtilis* mixture.") and corrected the y-axis label.

o B. subtilis should be included in panel c for comparison.

Reply: We would like to thank the reviewer's kind suggestion. We added *B. subtilis* data into **panel c** for comparison.

Figure 2c, Distribution of gene count of different bacterial species datasets by smRandom-seq, including *B. subtilis*, *E. coli*, *A. baumannii*, *K. pneumoniae*, *P. aeruginosa*, and *S. aureus*. The cell numbers of these bacterial species datasets by smRandom-seq were as follows: *B. subtilis* 45, *E. coli* 5001, *A. baumannii* 1105, *K. pneumoniae* 1127, *P. aeruginosa* 2973, and *S. aureus* 1989.

o If mention ncRNA's are detected, but they are not visible in the plot.

Reply: Thanks for the reviewer's careful reading. The proportion of ncRNA is too low to be observed in the percentage bar chart. We classified ncRNA and tRNA as other in the revised plot (**Fig. 2d**). We also provided a source data file (**Source Data Fig 2d**) of the proportions of biotype reads in **Fig. 2d**.

Figure 2d, Representative proportions of transcript categories of *E. coli* samples with (rRNA depletion) and without rRNA depletion (No rRNA depletion). other, all other RNA classes.

o The axis labels of panel f are incomplete.

Reply: Thanks for the reviewer’s careful reading. We completed the axis labels of **panel f** with “Gene expression (Log10(Counts+1)) of no rRNA depletion” and “Gene expression (Log10(Counts+1)) of rRNA depletion” in the revised **Fig. 2**.

Figure 2f, Correlation of gene expression (Log10(Counts+1)) of *E. coli* samples with and without rRNA depletion.

o E1-E4 in panel j should be explained as well as arrows above and next to the violin plots. E1-E4 should also be explained in the text, accordingly. The box plot is missing; instead, medians and standard deviations are shown.

Reply: We would like to thank the reviewer’s kind suggestions. We explained E1-E4 and arrows above and next to the violin plots in **panel j** in the figure legends (**Page 32**). We also explained E1-E4 in the results and method section, accordingly (**Page 11, 15**). E1, E2, E3, and E4 represent the GFP-*E. coli* samples induced with 0, 6.25, 25, and 100 mM propionate, respectively. Arrows to the left above the violin plots represent that the violin plots use the left y-axis. Arrows to the right above the box plots represent that the box plots use the right y-axis. We added the box plot in panel j.

Figure 2j, Comparison of qPCR relative expression (n=3) and UMI count of *E. coli* samples with a progressive increase in expression of GFP. E1, E2, E3, and E4 represent the GFP-*E. coli* samples induced with 0, 6.25, 25, and 100 mM propionate, respectively. Arrows to the left above the violin plots represent that the violin plots use the left y-axis. Arrows to the right above the box plots represent that the box plots use the right y-axis.

o Axis labels should be harmonized throughout the entire figure. E.g., gene count or gene counts; number of UMIs or median UMIs.

Reply: Thanks for the reviewer’s careful reading. We harmonized the axis labels throughout the

entire figure with “gene count” and “UMI count”.

o It is not clear how many single-cells of E. coli were analyzed for the individual panels (e.g. panel d, e, f...). This should be clarified in the figure as well as in the referring text.

Reply: We thank the reviewer for the valuable comment. We added the number of single cells analyzed for the individual panels in figure (**Fig. 2e, Fig. 2f, Fig. 2h**), as well as in the referring figure legends. The cell number of the *E. coli* samples with and without rRNA depletion used in **Fig. 2d, e, f, g, h** was 7645.

Figure 2d, Representative proportions of transcript categories of *E. coli* samples with (rRNA depletion) and without rRNA depletion (No rRNA depletion). other, all other RNA classes. The cell number of the *E. coli* samples with and without rRNA depletion used in **Fig. 2d, e, f** was 7645. **e**, Scatter plot showing gene count and the number of reads per barcode of *E. coli* samples with (blue) and without (red) rRNA depletion. Each dot corresponds to a putative cell. **f**, Correlation of gene expression ($\text{Log}_{10}(\text{Counts}+1)$) of *E. coli* samples with and without rRNA depletion. **g**, Gene count versus barcode rank plot. Identification of the barcodes that represent actual bacteria. The *E. coli* sample with rRNA depletion was sequenced more deeply for the next performance evaluations. Barcodes of the deeper *E. coli* sample with rRNA depletion are ordered from the largest to smallest gene count. On the gene count versus barcode rank plot, the knee point (red dashed line) indicates the threshold for actual *E. coli* sample (Cell number: 7645). **h**, Median gene count when down sampling total number of reads from the deeper *E. coli* sample with rRNA depletion to the indicated depths. The cell number of the deeper *E. coli* sample with rRNA depletion was 7645.

o 2k - it seems strange due to the heterogeneous nature of cells that they cluster like this - it's almost too perfect. We'd expect to at least see some sub clusters evident. How was the data transformed prior to this - this would greatly help.

Reply: We thank the reviewer for the valuable comment. We used two repeated exponential-phase *E. coli* samples from the same culture tube for the batch-effect experiment. Thus, these two samples were fairly homogeneous. We revised the description of the samples used for the batch experiment (**Page 8**): “two repeated exponential-phase *E. coli* samples from the same culture tube.” For the batch effect analysis in **Figure 2k**, we used the typical workflow of scRNA-seq data analysis. The “pandas.merge” function was used to create a data frame of the two batches without changing the expression matrix. The “scanpy.pp.normalize_total” function was used to normalize each cell by

total counts over all genes, so that every cell of the two batches has the same total count after normalization. The normalized datasets of the two batches in this paper displayed a proper alignment visualizing with UMAP dimensionality reduction (**Fig. 2k**) and a high correlation on gene expression ($R = 0.95$, $p < 2.2e-16$) (**Supplementary Fig. 5b**).

3.3 Figure 3:

o Panel d is missing a y-axis label (expression levels).

Reply: Thanks for the reviewer's careful reading. We added the y-axis labels of **panel d**.

o In panel f it is not clear what the gene ratio exactly means and what counts are. Further details in the figure legend would be helpful. Besides, top three counts are cropped.

Reply: Thanks for the reviewer's careful reading. We added further details in the figure legend of **panel f (Page 33)** and revised the **panel f** figure. Gene count is the number of DEGs enriched in a GO term. Gene ratio is the number of observed DEGs divided by the number of expected genes from each GO term.

o It is not clear to what the RNA-seq data shown in panel e, g, h was normalized to.

Reply: Thanks for the reviewer's careful reading. The scRNA-seq data shown in **panel e, g, h** was normalized using the typical workflow of scRNA-seq data analysis (**Page 21**). The "sc.pp.normalize_total" function normalizes each cell by total counts over all genes. Every cell has the same total count after normalization.

3.4 Figure 4:

o Cluster numbers should be more clearly highlighted and adjusted properly in panel c. Especially subclusters 2 and 4. Further, it is not clear what "leiden" means above.

Reply: Thanks for the reviewer's careful reading. We highlighted and adjusted the cluster numbers in **panel c**. Leiden is a modularity optimization algorithm in Scanpy used for clustering cells. We replaced it with "sub-clusters" in the revised **panel c**.

Figure 4c, UMAP projection of all the cells collected at the different time points, based on their gene expression colored by sub-clusters.

o Label of x-axis is missing in panel d.

Reply: Thanks for the reviewer’s careful reading. We added the y-axis labels of **panel d**.

Figure 4d, The correlation matrix showed the correlation coefficients between sub-clusters.

o The color code for the mean expression is missing in panel e and the data obtained in this panel should be better described in the results section.

Reply: Thanks for the reviewer’s careful reading. We added the color code for the mean expression in **panel e** and further described the data obtained in **panel e** in the results section (**Page 10**): “Therefore, we ranked genes for characterizing these subclusters and identified the top 5 DEGs to further understand the heterogeneity of *E. coli* upon AMP treatment (**Fig. 4e**). We selected these antibiotic response related DEGs (including *aceA*, *aceB*, *osmY*, *degP*, *fusA*, *ssrA*, *glpD*, *glpK*, *sulA*, *recA*, *fhuA*, and *secY*) representing each subcluster (**Fig. 4e**).”.

Figure 4e, Top 5 DEGs among subclusters.

o The scaling of figures f-m should be adjusted to allow direct comparison. Further the information about normalization is missing and it is not clear why the expression levels are even negative in some panels.

Reply: Thanks for the reviewer’s careful reading. We added the information about normalization and scaling of figures f-m in the Method section (**Page 21**). The “scanpy.pp.normalize_total” function with default parameters (target_sum=1e4) was used to normalize each cell by total counts over all genes. The “scanpy.pp.log1p” function with default parameters was used to logarithmize the data matrix and the “sc.pp.scale” function with default parameters (max_value=10) was used to scale each gene to unit variance and zero mean. After the scale function in each gene, the expression levels are incomparable among genes and the expression levels are negative in some panels.

o The labeling is off for panel k-m. Panel m is supposed to be a t-SNE plot according to the figure legend and not a UMAP for the cell division marker gene *ftsN*.

Reply: Thanks for the reviewer’s careful reading. We added the labeling for **panel k-m**. We corrected the figure legend of **panel m**, which is a UMAP plot (**Page 35**).

Figure 4f-l, Expression of glycerol flux related marker gene of subclusters 0 and 7 (*glpK*) (**f**), glyoxylates shunt related marker gene of subclusters 1, 3 and 4 (*aceA*) (**g**), periplasm related marker gene of subcluster 8 (*osmY*) (**h**), elongation factor G related marker gene of subcluster 2 (*fusA*) (**i**), protein quality control related marker gene of subcluster 5 (*ssrA*) (**j**), membrane related marker genes of subclusters 6 and 9 (*fhuA*, *secY*, respectively) (**k**), two DNA damage related marker genes of subcluster 10 (*recA*, *sulA*) (**l**) overlaid on the UMAP plot. Dashed red circle: the distinct sub-cluster 10. **m**. Expression of cell division related gene (*ftsN*) overlaid on the UMAP plot. Dashed red circle: the distinct sub-cluster 10.

o Dashed red lines used in panel l-o should be explained in the figure legend. Subcluster 6 should not be highlighted in the panel or according to the results section.

Reply: Thanks for the reviewer’s careful reading. We explained the dashed red lines used in **panel l-o** in the revised figure legend (**Page 35**). The dashed red circles in **panel l** and **m** represent the distinct sub-cluster 10. The dashed red rectangles in **panel n** and **o** represent the distinct gene expression patterns of sub-clusters 9 and 10, which are associated to 4h time point. Subcluster 6 was not highlighted in the revised **Figure 4**.

4. Discussion

4.1 “...which are usually not of interest.”

Reply: Thanks for the reviewer’s careful reading. We corrected this mistake in the revised manuscript (**Page 13**).

4.2 The mentioned issue with the template-switching strategy should be supported by suitable references.

Reply: We would like to thank the reviewer’s kind suggestion. We added a reference²¹ to support the mentioned issue with the template-switching strategy (**Page 12**).

4.3 The authors comment on the integration of a pre-indexing step. This should be explained in more

details how this could be technically solved and integrated. Addressed advantages would be the option to process multiple samples and the reduction of batch-effects. Though the data in figure 2 showed a clear limitation in sample throughput but at the same time no batch effects at all.

Reply: We thank the reviewer for the valuable comment. We provided details about how pre-indexing step could be technically solved and integrated with smRandom-seq in the revised manuscript (**Page 13**): “A pre-index strategy could be easily combined with smRandom-seq by adding indexes to RT primers according to the published scifi-RNA-seq²², where the bacteria were split into different tubes for RT with pre-indexed random primers, and then those pre-indexed samples could be combined into one sample for the following barcoding procedures.”. We revised the advantage of pre-indexing (**Page 13**). The major advantage of pre-indexing is to increase the throughput. As mentioned above, the throughput of current smRandom-seq is ~10,000. By adding a pre-indexing step with 96 unique pre-indexed RT primers, the throughput of smRandom-seq can be expanded to ~1,000,000 cells.

5. Material and Methods

5.1 Details about the used reverse transcriptase kit should be given.

Reply: We would like to thank the reviewer’s kind suggestion. We add the details about the used reverse transcriptase kit (**Page 16**): “The reverse transcriptase kit (Cat. R20114124), including reverse transcriptase (50 U/μL), 5× reverse transcription buffer, and dNTP Mix (100 mM), was ordered from M20 Genomics. The RT reaction mix (per 50-μL reaction) was prepared on ice, including a few millions of bacteria in 27.5 μL PBS, 10 μL 5× reverse transcription buffer, 5 μL 10 μM random primer (**sequence in Supplementary Table 1**), 2.5 μL 100 mM dNTP, 2.5 μL RNase inhibitor, 2.5 μL reverse transcriptase (50 U/μL), and incubated with twelve cycles of multiple annealing ramping from 8°C to 42°C, followed by a 42 °C 30 minutes, in a thermal cycler.”

5.2 Further details should be given for the barcode beads synthesis, especially as this was a customized and adapted step compared to previous work.

Reply: We would like to thank the reviewer’s kind suggestion. We added further details for the barcode beads synthesis in the revised manuscript (**Page 17, 18**), including the customized and adapted steps compared to previous work.

5.3 The information about the used ratio of the cDNA: Cas9 enzyme: sgRNA should be added.

Reply: Thanks for the reviewer’s careful reading. We added ratio of the cDNA: Cas9 enzyme: sgRNA (1: 100: 1000) in the revised manuscript (**Page 19**).

5.4 Details about the used reference genomes and annotation files for the data analysis are missing.

Reply: Thanks for the reviewer’s careful reading. We added The GenBank assembly accession of reference genomes and annotation files used in this study in the revised manuscript (**Page 20**): “The GenBank assembly accession of reference genomes and annotation files used in this study are as

follows: GCF_902728005.1 for *A. baumannii*; GCF_000750555.1 for *E. coli*, GCF_000775955.1 for *K. pneumoniae*, GCA_000006765 for *P. aeruginosa*, GCF_000737615.1 for *S. aureus*, GCF_002055965.1 for *Bacillus subtilis*.”

5.5 “*annotations were performed using featureCounts*” - *I’m not sure featureCounts annotates. It counts reads from an alignment file such as a .bam or .sam file.*

Reply: Thanks for the reviewer’s careful reading. We revised the description of featureCounts (**Page 20**): “Read summarization was performed using the featureCounts (2.0.1)¹¹. FeatureCounts takes the GTF format annotation file and the alignment file (a .bam file) as input to assign mapped reads to genomic features, and then counts reads. The GenBank assembly accession of GTF format annotation files were provided above.”

5.6 *You mention the tool Origin that was used for data analysis and graphing - there is no reference so we can’t determine the specific tool – we assume this was just used with graphing. Can you comment on why this tool was used instead of R, which was also clearly used.*

Reply: Thanks for the reviewer’s careful reading. We added the version of Origin (OriginPro 2021 (9.8.0.200)). We used Origin for convenience to analyze data and graph **Fig. 2d, h, j**.

6. Supplementary Material

6.1 *Supplementary Fig. 3: We assume that the data shown in panel a and b were obtained without rRNA depletion. However, it is not fully clear and should be clarified in the figure legend. Further, the authors should comment why they detected much higher numbers of genes in B. subtilis compared to E. coli as well as to all other tested species in Fig. 2c.*

Reply: We thank the reviewer for the comments. We revised the figure legend of **supplementary Fig. 3**: “**a, b**, Distribution of UMI counts (**a**) and gene (**b**) counts detected from *B. subtilis* and *E. coli* in the mixture of *B. subtilis* and *E. coli* sample by smRandom-seq without rRNA depletion.”. We also agree with that there was bias in species-mixing experiment by smRandom-seq due to the variations in bacterial size, RNA content and cell wall composition. We are working on reducing the bias in smRandom-seq so that it can be used in microbiome. We added the comment on the bias in the revised manuscript (**Page 6**): “However, the current smRandom-seq used in species-mixing experiments showed bias in cell number, UMI count, and gene count due to variations in bacterial size, RNA content and cell wall composition.”

6.2 *Axis labels in Supplementary Fig. 5b are incomplete.*

Reply: We thank for the reviewer's suggestion. We completed the axis labels in Supplementary Fig. 5b.

Supplementary Figure 9b (Supplementary Fig. 5b), Scatter plot showing correlation of gene expression (Log10(Counts+1)) for two repeated *E. coli* samples (Batch1 and Batch2) applied with smRandom-seq separately.

6.3 Panel a can be skipped in Supplementary Fig. 6 as there is only a single panel. Same applies for the text. Consider changing the x-axis label to minutes or hours.

Reply: Thanks for the reviewer's careful reading. We skipped panel a in Supplementary Fig. 6 and the revised text. We changed the labels to hours on x-axis.

Supplementary Figure 10 (Supplementary Fig. 6). Growth curves (OD 600) of *E. coli* upon 0, 6.25, 12.5, 25, and 50 µg/mL CIP.

6.4 Supplementary Fig. 8 b includes still underline from spell check.

Reply: Thanks for the reviewer's careful reading. We removed the underlines in **Supplementary Figure 12b** (Supplementary Fig. 8 b).

Supplementary Figure 12b (Supplementary Fig. 8 b), Image of *E. coli* samples treated with 5 µg/mL AMP after 0, 1, 2, and 4 hours. Scale bar: 10 µm.

6.5 It is not clear where in the text/figures some of the supplementary tables are linked to. The supplementary tables mentioned in the text should also be labeled accordingly.

Reply: We thank for the reviewer's suggestion. We added the labels of supplementary tables in the revised manuscript (**Page 28**): "Supplementary Table 1 List of primers used in smRandom-seq, including Barcoded beads primer for hydrogel beads synthesis (**Supplementary Fig. 2c**), Random primer for RT (**Fig. 1**), PCR-1 primer and PCR-2 primer for PCR amplification (**Fig. 1**).

Supplementary Table 2 Barcode sequences used in hydrogel beads synthesis of smRandom-seq (**Supplementary Fig. 2c**).

Supplementary Table 3 sgRNAs for rRNA depletion of smRandom-seq (**Fig. 2d, e, f, Supplementary Fig. 6**).

Supplementary Table 4: A summary of the datasets by smRandom-seq (**Fig. 2, 3, 4**)"

Reviewer #3:

This paper reports a droplet-based method for scRNA-seq of bacteria. The results look promising but not enough information is provided to put them into context.

Reply: We are grateful to reviewer's effort to review our paper. We have fully addressed the concerns as follows.

1. Several important references are missing from the manuscript. First, there already are high throughput methods for scRNA-seq of bacteria that use some form of droplet microfluidics. Second, there are at least two reports that apply rRNA depletion in the context of bacterial scRNA-seq. These papers need to be referenced and appropriately compared to:

Droplet based scRNA-seq for bacteria:

- McNulty *et al.* Droplet-based single cell RNA sequencing of bacteria identifies known and previously unseen cellular states

<https://www.biorxiv.org/content/10.1101/2021.03.10.434868v2.abstract>

- Wang *et al.* "Massively-parallel Microbial mRNA Sequencing (M3-Seq) reveals heterogeneous behaviors in bacteria at single-cell resolution"

<https://www.biorxiv.org/content/10.1101/2022.09.21.508688v2>

- Ma *et al.* "Bacterial droplet-based single-cell RNA-seq reveals heterogeneity in bacterial populations and in response to antibiotic perturbation"

<https://www.biorxiv.org/content/10.1101/2022.08.01.502326v1.abstract>

There are two papers that combine rRNA depletion with bacterial scRNA-seq:

- Homberger *et al.* "Improved bacterial single-cell RNA-seq through automated MATQ-seq and Cas9-based removal of rRNA reads"

<https://www.biorxiv.org/content/10.1101/2022.11.28.518171v1.full.pdf>

- Wang *et al.* "Massively-parallel Microbial mRNA Sequencing (M3-Seq) reveals heterogeneous behaviors in bacteria at single-cell resolution"

<https://www.biorxiv.org/content/10.1101/2022.09.21.508688v2>

Reply: We thank the reviewer for the valuable comment. We noticed that Ma *et al.*'s paper has been just published in *Cell*²³. We updated the reference for this paper. We added these references and compared these methods with smRandom-seq in the revised manuscript (**Page 13**). These three bacterial single cell RNA-seq methods, McNulty *et al.*'s method²⁴, M3-Seq²⁵, and BacDrop²³ were reported using the commercially 10X Genomics microfluidic single cell sequencing platform. To efficiently barcode the extremely low content RNAs in single bacterium by smRandom-seq, we optimized our previous droplet barcoding platform designed for single cell barcoding⁵, by generating smaller barcoded beads and droplets, and modifying the primers releasing and the barcodes synthesis strategy.

We also compared the rRNA-depletion methods of these two papers with smRandom-seq (**Page 13**). Both Homberger *et al.*'s method⁸ and smRandom-seq used the Cas9-based rRNA depletion strategy. However, Homberger *et al.* developed a low-throughput improved bacterial MATQ-seq. Wang *et al.*

reported another post-hoc rRNA depletion scheme using a duplex-specific nuclease to remove the rRNAs hybridized with DNA probes targeting rRNAs of specific bacterial species.

2. *The data analysis section needs to be expanded and analysis code should be provided on GitHub or similar. For example, authors need to include information about which genomes were used for alignment, how multiply-aligning reads were handled, how down-sampling analysis was performed etc.*

Reply: We thank for the reviewer's kind suggestions. We expanded the data analysis section including information about which genomes were used for alignment, how multiply-aligning reads were handled, how down-sampling analysis was performed (**Page 20, 21**): “The GenBank assembly accession of reference genomes and annotation files used in this study are as follows: GCF_902728005.1 for *A. baumannii*; GCF_000750555.1 for *E. coli*, GCF_000775955.1 for *K. pneumoniae*, GCA_000006765 for *P. aeruginosa*, GCF_000737615.1 *S. aureus*, GCF_002055965.1 for *Bacillus subtilis*. Only uniquely mapped reads were used for analyses. Sequencing reads were randomly subsampled to 22k, 18k, 13k, 9k, 4k, 2k, 1k reads per cell, and the analysis was repeated to calculate the median numbers of genes detected per cell.” We also added a supporting link to GitHub in the revised manuscript (**Page 22**): “All script files used in the analysis in this manuscript can be downloaded from GitHub at are available at <https://github.com/wanglab2023/Msc-RNA-seq>.”

3. *Minor: PETRI-seq and microSPLiT-seq also use random reverse transcription primers and perform RT inside fixed cells.*

Reply: Thanks for the reviewer’s careful reading. We added these characters of PETRI-seq and microSPLiT-seq in the revised manuscript (**Page 3**): “Two previously developed bacterial scRNA-seq methods, PETRI-seq³ and microSPLiT⁴, using random reverse transcription (RT) primers, enable researchers to analyze thousands of fixed bacteria simultaneously with a split-pool barcoding strategy¹⁹,”

4. *My biggest concern is around the reporting of gene and UMI counts. There are several places in the paper where the authors claim that they detect on the order of 1000 genes per cell. (E.g. “A median of ~1000 genes can be detected per bacterium in a population of ~8000 *E. coli*.”) However, it is not entirely clear whether this is an actual measurement or a computational extrapolation. If the authors indeed (experimentally) detect 1000 genes/cell that would be a great improvement over the state of the art. However, the experimental data in Fig. 2C shows a median gene count of 225 which is much more modest but would be in line with gene counts reported in earlier bacterial scRNA seq papers. Please make it clear which numbers are measurements and which are extrapolation.*

Reply: We thank the reviewer for the valuable comment. The performance of ~1000 median genes per cell by smRandom-seq is an actual measurement, which was achieved by combining a rRNA-depletion step and deeper sequencing depth (~50K raw reads per cell) (**Fig. 2h**). The *E. coli* data (a median gene count of 225) in **Fig. 2c** was generated by the smRandom-seq without rRNA-depletion

and shallower sequencing depth. We provided a summary (**Supplementary table 4**) of the datasets by smRandom-seq including mean total raw reads per cell, mean total mapped reads per cell, and median gene count per cell.

5. Fig. 2i lists 1045 genes detected/cell for *E. coli* at ~50K reads but Fig. 2h shows 1045 genes detected at ~22K reads.

Reply: Thanks for the reviewer's careful reading. The "~50K reads" in **Fig. 2i** were the number of raw reads and the "~22K reads" in **Fig. 2h** was the number of mapped reads. PETRI-seq³ and microSPLiT⁴ reported the number of raw reads in the published papers, thus we compared the number of raw reads in **Fig. 2i**. We provided the mean total raw reads per cell and mean total mapped reads per cell of all the datasets by smRandom-seq in a summary (**Supplementary table 4**).

6. Fig. 2h caption says "Number of genes detected per *E. coli* when down sampling total read counts to the indicated depths." Please provide a description of how this down sampling analysis is performed and which experimental data were used.

Reply: We thank for the reviewer's kind suggestion. We provided a description of how this down sampling analysis is performed and which experimental data were used (**Page 20**): "The deeper sequenced *E. coli* data with rRNA-depletion were used to generate the saturation curve. Sequencing reads were randomly subsampled to 22k, 18k, 13k, 9k, 4k, 2k, 1k reads per cell, and the analysis was repeated to calculate the median numbers of genes detected per cell."

7. For a different experiment the paper reports "We detected a median count of 6564 and 1785 UMIs per cell for *B. subtilis* and *E. coli*, as well as a median count of 1249 and 429 genes per cell, respectively (Supplementary Fig. 3a, b)." Please report rRNA and mRNA UMIs separately. Also, given that the *E. coli* gene count is about twice that reported in Fig. 2C, is this a result of differences in sequencing depth? How many cells were used in that experiment?

Reply: We thank the reviewer for the valuable comments. We provided the rRNA and mRNA UMIs of the *B. subtilis* and *E. coli* separately (**Supplementary Fig. 3c, d**). The results showed that the UMI count of rRNA was relatively low compared to that of mRNA. rRNA-derived reads were usually multiply mapped. In our data analysis, only unique mapped reads were used for the downstream analysis. Thus, most of the rRNA-derived reads were dropped in the mapping step. The sequencing depth of *E. coli* samples in the two-species mixture experiment (**Fig. 2a, b**) was higher than that of the single-species experiment (**Fig. 2c**). The mean total mapped reads per cell of these two experiments were provided in the summary (**Supplementary table 4**). The total cell number used in the species-mixing experiment was ~500.

Figure 3c, d, Distribution of mRNA UMI counts (**c**) and rRNA UMI (**d**) counts detected from *B. subtilis* and *E. coli* in the mixture of *B. subtilis* and *E. coli* sample by smRandom-seq without rRNA depletion.

8. Are the data in Fig. 2e from a different experiment than those in 2C? The UMI counts (before depletion) are different.

Reply: Thanks for the reviewer's careful reading. The *E. coli* data (before rRNA depletion) in **Fig. 2e** and the *E. coli* data in **Fig. 2c** are from the same experiment. We identified 5001 putative cells from this *E. coli* data without rRNA depletion in **Fig. 2c**. The median gene count of these 5001 cells in **Fig. 2c** was 225. The median UMI count of these 5001 putative cells in **Fig. 2c** was 428 (showed in **Supplementary Fig. 5a**). Next, the *E. coli* library was applied with rRNA depletion, and we identified 7645 putative cells in the rRNA-depleted *E. coli* data in **Fig. 2e**. To fairly compare the sensitivity of smRandom-seq methods with or without rRNA depletion, we selected more putative cell barcodes (7645) from the Gene count vs Barcode plot of the previous *E. coli* data in **Fig. 2c**. Thus, the median gene count and UMI count of this *E. coli* data (before rRNA depletion) in **Fig. 2e** was 192 and 332, respectively, which is slightly less than that in **Fig. 2c**.

9. Fig. 2e and Supplementary fig. 3c: Does each dot correspond to a barcode? is a barcode a putative cell?

Reply: Thanks for the reviewer's careful reading. Yes. Each dot corresponds to barcode. A barcode is a putative cell in **Fig. 2e** and **Supplementary fig. 6a** (**Supplementary fig. 3c**).

10. Please provide a summary of the experiments performed including the number of cells, the number of raw sequencing reads per cell, the number or mRNA UMIs, rRNA UMIs, genes etc.

Reply: We thank for the reviewer's the valuable suggestion. We provided a summary (**Supplementary table 4**) of the experiments performed by smRandom-seq including the cell numbers, thresholds, reads numbers, UMI counts and gene counts for different biotypes.

11. Minor: What is the purpose of comparing cell numbers in Fig. 2i? Both PETRI-seq and microSPLiT were used to perform experiments with many more cells than listed.

Reply: Thanks for the reviewer's careful reading. To compare the sequencing sensitivity of smRandom-seq with that of other methods, we chose the *E. coli* datasets with existing performance information (including median genes, median operons, proportion of rRNA and mRNA) in the published papers of PETRI-seq³ (exponential-phase *E. coli* dataset from experiment 2.01) and microSPLIT⁴ (*E. coli* dataset from the *E. coli* and *B. subtilis* species-mixing experiment). The throughputs of these two methods and smRandom-seq are all ~10,000 cells. Cell numbers can influence sequencing data volume and sequencing depth. Thus, we take the cell numbers into consideration in **Fig. 2i**.

12. *Fig. 1a implies that multiple UMIs could be derived from the same transcript.*

Reply: We thank the reviewer for the valuable comment. Yes, multiple UMIs could be derived from the same transcript. Thus, we performed the GFP expression model (**Figure 2j, Supplementary Figure 8**) to evaluate the accuracy of gene expressions by smRandom-seq. Additionally, we normalize each cell by total counts over all genes before bioinformatic analysis.

Reference

1. Mathur, L. et al. Combi-seq for multiplexed transcriptome-based profiling of drug combinations using deterministic barcoding in single-cell droplets. *Nat Commun* **13**, 4450 (2022).
2. Zilionis, R. et al. Single-cell barcoding and sequencing using droplet microfluidics. *Nat. Protoc.* **12**, 44-73 (2017).
3. Blattman, S.B., Jiang, W., Oikonomou, P. & Tavazoie, S. Prokaryotic single-cell RNA sequencing by in situ combinatorial indexing. *Nat Microbiol* **5**, 1192-1201 (2020).
4. Kuchina, A. et al. Microbial single-cell RNA sequencing by split-pool barcoding. *Science* **371** (2021).
5. Klein, Allon M. et al. Droplet Barcoding for Single-Cell Transcriptomics Applied to Embryonic Stem Cells. *Cell* **161**, 1187-1201 (2015).
6. Macosko, Evan Z. et al. Highly Parallel Genome-wide Expression Profiling of Individual Cells Using Nanoliter Droplets. *Cell* **161**, 1202-1214 (2015).
7. Zheng, G.X. et al. Massively parallel digital transcriptional profiling of single cells. *Nat Commun* **8**, 14049 (2017).
8. Homberger, C., Hayward, R.J., Barquist, L. & Vogel, J. Improved bacterial single-cell RNA-seq through automated MATQ-seq and Cas9-based removal of rRNA reads. *bioRxiv*, 2022.2011.2028.518171 (2022).
9. Martin, M. Cutadapt removes adapter sequences from high-throughput sequencing reads. *2011* **17**, 3 (2011).
10. Dobin, A. et al. STAR: ultrafast universal RNA-seq aligner. *Bioinformatics* **29**, 15-21 (2013).
11. Liao, Y., Smyth, G.K. & Shi, W. featureCounts: an efficient general purpose program for assigning sequence reads to genomic features. *Bioinformatics* **30**, 923-930 (2013).
12. Smith, T., Heger, A. & Sudbery, I. UMI-tools: modeling sequencing errors in Unique Molecular Identifiers to improve quantification accuracy. *Genome Res.* **27**, 491-499 (2017).
13. Wolf, F.A., Angerer, P. & Theis, F.J. SCANPY: large-scale single-cell gene expression data analysis. *Genome Biol.* **19**, 15 (2018).

14. Method of the Year 2013. *Nat. Methods* **11**, 1-1 (2014).
15. Papalexi, E. & Satija, R. Single-cell RNA sequencing to explore immune cell heterogeneity. *Nat. Rev. Immunol.* **18**, 35-45 (2018).
16. Suvà, M.L. & Tirosh, I. Single-Cell RNA Sequencing in Cancer: Lessons Learned and Emerging Challenges. *Mol. Cell* **75**, 7-12 (2019).
17. Armand, E.J., Li, J., Xie, F., Luo, C. & Mukamel, E.A. Single-Cell Sequencing of Brain Cell Transcriptomes and Epigenomes. *Neuron* **109**, 11-26 (2021).
18. Giannoukos, G. et al. Efficient and robust RNA-seq process for cultured bacteria and complex community transcriptomes. *Genome Biol.* **13**, R23 (2012).
19. Rosenberg, A.B. et al. Single-cell profiling of the developing mouse brain and spinal cord with split-pool barcoding. *Science* **360**, 176-182 (2018).
20. Blázquez, J., Rodríguez-Beltrán, J. & Matic, I. Antibiotic-Induced Genetic Variation: How It Arises and How It Can Be Prevented. *Annu. Rev. Microbiol.* **72**, 209-230 (2018).
21. Sheng, K., Cao, W., Niu, Y., Deng, Q. & Zong, C. Effective detection of variation in single-cell transcriptomes using MATQ-seq. *Nat. Methods* **14**, 267-270 (2017).
22. Datlinger, P. et al. Ultra-high-throughput single-cell RNA sequencing and perturbation screening with combinatorial fluidic indexing. *Nat. Methods* **18**, 635-642 (2021).
23. Ma, P. et al. Bacterial droplet-based single-cell RNA-seq reveals antibiotic-associated heterogeneous cellular states. *Cell* **186**, 877-891.e814 (2023).
24. McNulty, R., Sritharan, D., Liu, S., Hormoz, S. & Rosenthal, A.Z. Droplet-based single cell RNA sequencing of bacteria identifies known and previously unseen cellular states. *bioRxiv*, 2021.2003.2010.434868 (2021).
25. Wang, B. et al. Massively-parallel Microbial mRNA Sequencing (M3-Seq) reveals heterogeneous behaviors in bacteria at single-cell resolution. *bioRxiv*, 2022.2009.2021.508688 (2022).

Reviewers' Comments:

Reviewer #1:

Remarks to the Author:

In the new version of the manuscript, Xu et al. addressed all points that I had raised very comprehensively. For example, instead of just providing a UMAP plot of existing sequencing data (Fig. 2C, which was not really suited for this kind of analysis due to separate sampling and potential batch effects) they also included new experimental data of a species mixing experiment (new Fig. 4). Similarly, the new results comparing functional readouts with transcriptomic data seem convincing. I also very much appreciate the many new data sets that were provided upon request of the other reviewers. From my side, the manuscript is now ready for publication in Nature Communications.

Reviewer #2:

Remarks to the Author:

We would like to thank the authors for addressing most of the previous issues and their willingness to provide additional files and underlying code. We believe that a revised version that also addresses the remaining remarks listed below, should be ready for publication.

Specific comments

- We have noticed that some of the preprints have now been published such as ref #45, perhaps this would be a good opportunity to double check these.
- Regarding our comment that the figures still "require some rework with a focus on layout and specificity", I highly recommend the authors to go through again carefully as there are still some typos as mentioned before. For example, Fig. 2 : A. baumannii (missing the double n); depletion (spelt incorrectly). In addition, figure labels are often squished (e.g. Fig. 1 d-h; Fig. 2 d) and labels should be harmonised to allow a direct comparison (e.g. Fig. 3d; 4 f-m). In the current version it can be difficult for the reader to read and grasp the individual panels.
- Some statistical thresholds remain to be clarified before the reader can confidently interpret some of the results. For example, what thresholds were used to determine marker genes, DEG, and also for GO enrichment – q-value, p-value? < 0.05 or < 0.01? At the moment, there is the assumption that the false discovery rate has been corrected for and the thresholds are statistically relevant.
- p.15 in manuscript: Species names should be in italics.
- The code on GitHub has now been provided, but this code has no comments and would be very challenging to reproduce. Ideally the readme file will have step by step instructions of which script to run first and what the output will be.
- Authors may also need to double check some of the code that everything from the manuscript is actually there. For example, the STAR script does not include the genome index step. The r-script and python scripts have the same issue, no comments and no description of what the code is doing. This is needed for reproducibility.
- The code for the downsampling procedure is either missing or impossible to find due to the lack of comments.
- Authors have used use gene count in various places in the results, but in some areas of the methods they have also used detected genes to describe the same thing. It's important to keep terms consistent across studies and suggest changing to detected genes to keep in line with this.

Figure 2:

- As discussed above, updating the y-axis to detected genes would be helpful. In addition, changing the word barcode to cells would also help make this plot to be more easily understandable.
- The authors now provide more information on the data analysis, however, more details on parameters and cut-offs should be given on the downsampling experiment performed in panel h.
- Panel j, since you have the plot type on the primary and secondary y-axis, you can remove the arrows inside of the plot.
- As explained by the authors panel k was based on samples taken from the same subculture. Thus, the presented results provide information about the technical reproducibility but not about batch effects. If the focus were on batch effects, this will require independent replicates to compare those, such that confounding effects would indeed be introduced. Alternatively, just change the description of the figure to be about technical reproducibility.

Figure 3:

- Panel d, "expression levels" is too general. Assuming this is log2 (normalised counts)? If so, please update. Also applies to Figure 4.

Figure 4:

- Regarding panels f-m, why would a comparison between genes impossible? The scaling can be adjusted to include negative values, and using a common scale across plots like you are showing is common practice. All these plots should use the same scale.

Methods:

- For the identification of cell barcodes, authors write they used "reasonable parameters". This is very unspecific. Why not provide the actual parameters?
- Saturation analysis – how were they randomly sub-sampled? Also, the saturation section should be moved to after "Analysis of gene expression matrix".

Supplementary Figure 7:

- Thanks for including these two plots. The right-hand side plot is very informative and highlights the 5'-3' read coverage of your method. Panel a, however, is not that informative due to the sparse nature of sequencing single cells, so it could be removed. Naturally, this decision is up to the authors.

Reviewer #3:

Remarks to the Author:

Overall, the authors have done a good job at addressing reviewer comments but I still have a major concern.

The method of this paper is clearly very sensitive and seems to result in very good UMI counts across many species. However, in some instances, the numbers reported are really extraordinarily high to the point that I find them difficult to believe without some additional analysis or discussion.

For B. subtitles the authors report detecting 6564 umis from 1249 genes in 45 cells. The UMI

numbers are about 10-20x higher than any previously reported. My concern is that these extremely high numbers are some sort of artifact resulting from cell clumps, very unusual growth conditions, PCR or data analysis. Please perform at least the following quality controls:

1. Supplementary Fig. 3 shows that the mRNA UMI counts in E.coli are $\sim 3x$ lower than in B. subtilis but that rRNA counts are 3x higher. Is it possible that some B. subtilis rRNA genes are incorrectly labeled as mRNA? Please carefully check gene annotations and functions.
2. Include a gene expression analysis to show which specific genes are most highly expressed and how many mRNA are detected for these genes. Please also consider correlating your gene expression data to previous bulk or single cell datasets for B. subtilis. While this analysis cannot address issues arising from cell clumping, it would alleviate concerns relating to the analysis pipeline or to unusual growth conditions.
3. Another observation is that the UMI and gene counts are generated from a very small number of 45 cells. What criterion was used to pick such a small number? Is there a graph similar to 2g for B. subtilis?

Another example that I find surprising is the data shown in Fig. S11 where E.coli UMIs > 2000 are shown which is more than even in the species mixing experiment. Again, it would be helpful to indicate which genes drive this expression. What are the top 10 expressed genes at $t=0$ and how do they compare to the other E. coli datasets in the same paper and prior work?

REVIEWER COMMENTS

Reviewer #1 (Remarks to the Author):

In the new version of the manuscript, Xu et al. addressed all points that I had raised very comprehensively. For example, instead of just providing a UMAP plot of existing sequencing data (Fig. 2C, which was not really suited for this kind of analysis due to separate sampling and potential batch effects) they also included new experimental data of a species mixing experiment (new Fig. 4). Similarly, the new results comparing functional readouts with transcriptomic data seem convincing. I also very much appreciate the many new data sets that were provided upon request of the other reviewers. From my side, the manuscript is now ready for publication in Nature Communications.

Reply: Thanks for the reviewer's comments.

Reviewer #2 (Remarks to the Author):

We would like to thank the authors for addressing most of the previous issues and their willingness to provide additional files and underlying code. We believe that a revised version that also addresses the remaining remarks listed below, should be ready for publication.

Reply: We are grateful to reviewer's effort to review our paper. We have fully addressed the concerns as follows.

Specific comments

- We have noticed that some of the preprints have now been published such as ref #45, perhaps this would be a good opportunity to double check these.

Reply: We thank for the reviewer's careful reading. We have updated references of the published preprints, including ref#43¹ and ref#47² in the latest references.

- Regarding our comment that the figures still "require some rework with a focus on layout and specificity", I highly recommend the authors to go through again carefully as there are still some typos as mentioned before. For example, Fig. 2 : *A. baumannii* (missing the double n); depletion (spelt incorrectly). In addition, figure labels are often squished (e.g. Fig. 1 d-h; Fig. 2 d) and labels should be harmonised to allow a direct comparison (e.g. Fig. 3d; 4 f-m). In the current version it can be difficult for the reader to read and grasp the individual panels.

Reply: Thanks for the reviewer's kind suggestions. We corrected the mistakes and adjusted the layout of figures. We harmonized the labels of **Fig. 3d**.

Figure 3d, Violin plot showed the expression levels of outer membrane protein encoding genes (*ompF*, *tsx*, and *lamB*) in samples.

We also tried to harmonize the labels of **Fig. 4f-m** to -0.5~5 showing as below. However, the UMAP scatter plots with harmonized the labels were not suitable for the visualizations of the different expression levels among single cells or subclusters. For example, the different expression levels of *gldK* and *aceA* among subclusters were indistinguishable in the UMAP scatter plots with the scale of -0.5~5. Thus, we did not make the change of the labels of **Fig. 4f-m**.

Figure 4f-m, Expression levels of marker genes.

- Some statistical thresholds remain to be clarified before the reader can confidently interpret some of the results. For example, what thresholds were used to determine marker genes, DEG, and also for GO enrichment – q -value, p -value? < 0.05 or < 0.01 ? At the moment, there is the assumption that the false discovery rate has been corrected for and the thresholds are statistically relevant.

Reply: Thanks for the reviewer’s kind suggestions. We added the thresholds used for marker genes, DEG, and GO enrichment in Method and figure legends sections.

(Page 21): “The marker gene was set to a p -value < 0.05 and ordered according to scores. The top 20 marker genes were retained as the DEGs.” “The “enrichGO” function of clusterProfiler R package (version 3.16.0) was used for GO enrichment analysis (p -value cutoff value= 0.05 and q -value cutoff value= 0.05) of the top 20 DEGs of each group in R studio (version 4.0.1).”

(Page 34): “The cutoff value of p -value of GO enrichment was 0.05 and the cutoff value of q -value of GO enrichment was 0.05.”

- p.15 in manuscript: Species names should be in italics.

Reply: We thank for the reviewer’s careful reading. We revised the typeface of species names in in manuscript (Page 15): “*Escherichia coli* BW25113 (*E. coli*), *Bacillus subtilis* NCBI3610 (*B. subtilis*), *Acinetobacter baumannii* ATCC17978 (*A. baumannii*), *Klebsiella pneumoniae* XH209 (*K. pneumoniae*), *Pseudomonas aeruginosa* PAO1 (*P. aeruginosa*) and *Staphylococcus aureus* subsp. *Aureus* SA268 (*S. aureus*) were provided by Sir Run Run Shaw Hospital, Zhejiang University School of Medicine.”

- The code on GitHub has now been provided, but this code has no comments and would be very challenging to reproduce. Ideally the readme file will have step by step instructions of which script to run first and what the output will be.

Reply: Thanks for the reviewer’s suggestion. We added the comments of the code on GitHub and added the step-by-step instructions in the readme file.

- Authors may also need to double check some of the code that everything from the manuscript is actually there. For example, the STAR script does not include the genome index step. The r-script and python scripts have the same issue, no comments and no description of what the code is doing. This is needed for reproducibility.

Reply: Thanks for the reviewer's valuable comments. We added the code for the STAR genome index step, saturation analysis, gene body coverage, correlation analysis and technical reproducibility analysis. We added the comments and description of the r-script and python scripts.

- The code for the downsampling procedure is either missing or impossible to find due to the lack of comments.

Reply: We thank for the reviewer's careful reading. We added the code (Saturation analysis.sh) for the downsampling procedure on GitHub.

- Authors have used use gene count in various places in the results, but in some areas of the methods they have also used detected genes to describe the same thing. It's important to keep terms consistent across studies and suggest changing to detected genes to keep in line with this.

Reply: Thanks for the reviewer's kind suggestion. We replaced "gene count" with "detected genes" in the revised manuscript.

Figure 2:

- As discussed above, updating the y-axis to detected genes would be helpful. In addition, changing the word barcode to cells would also help make this plot to be more easily understandable.

Reply: Thanks for the reviewer's kind suggestion. We updated the y-axis to detected genes in the revised **figure 2** and supplementary figures.

The **figure 2g** is a Gene count vs Barcode plot, which is widely used in the pre-processing of scRNA-seq data to identify putative cells from all possible barcodes according to the position where the number of detected genes drops sharply. The barcodes on left side of the threshold were identified as putative cells, but the barcodes on right side of the threshold were identified as noise. Thus, we did not change the word barcode to cells in **figure 2g**. We revised the figure legend of **figure 2g** to make this plot to be more easily understandable in the manuscript.

(Page 32, 33): "g. The barcode rank plot of \log_{10} (total number of detected genes per barcode) versus \log_{10} (Barcode number) for each possible barcode. On the barcode rank plot, the knee point (red dashed line) indicated the threshold for putative cell. The barcodes on left side of the threshold were identified as putative *E. coli* (Cell number: 7645), but the barcodes on right side of the threshold were identified as noise."

- The authors now provide more information on the data analysis, however, more details on

parameters and cut-offs should be given on the downsampling experiment performed in panel h.

Reply: Thanks for the reviewer's kind suggestion. We provided the details on the downsampling experiment in the Method and figure legend sections of the revised manuscript.

(Page 22): "**Saturation analysis**

The rRNA depleted and deeper sequenced *E. coli* sample were used to generate the saturation curve. Saturation curve plot was generated by randomly selecting the corresponding number of raw reads from the sample library and then using the same alignment pipeline to calculate the median numbers of genes detected per cell. Sequencing reads were randomly subsampled to 22k, 18k, 13k, 9k, 4k, 2k, 1k reads per cell using the Seqtk tool."

(Page 33): "**h**, Saturation curve for the rRNA depleted and deeper sequenced *E. coli* sample (Cell number: 7645). Each point on the curve represents the median of detected genes when down sampling total number of reads from the library to the indicated depths (22k, 18k, 13k, 9k, 4k, 2k, 1k reads per cell)."

- Panel j, since you have the plot type on the primary and secondary y-axis, you can remove the arrows inside of the plot.

Reply: Thanks for the reviewer's kind suggestion. We removed the arrows inside of **figure 2j**.

- As explained by the authors panel k was based on samples taken from the same subculture. Thus, the presented results provide information about the technical reproducibility but not about batch effects. If the focus were on batch effects, this will require independent replicates to compare those, such that confounding effects would indeed be introduced. Alternatively, just change the description of the figure to be about technical reproducibility.

Reply: Thanks for the reviewer's valuable comments. We have changed the description of **figure 2k** and **supplementary figure 9** to be about technical reproducibility in the revised manuscript (Page 8): "To prove the technical reproducibility of smRandom-seq," (Page 12): "We provided sufficient and systematic evidence to prove the efficiency and technical reproducibility of smRandom-seq and validated the performance of smRandom-seq by practical application."

We also renamed the two samples in **figure 2k** and **supplementary figure 9** using Repeat 1 and Repeat 2. (Page 33): "**k**, UMAP projection of two repeated *E. coli* samples (Repeat 1 and Repeat 2) applied with smRandom-seq separately."

Figure 2k, UMAP projection of two repeated *E. coli* samples (Repeat 1 and Repeat 2) applied with smRandom-seq separately.

Figure 3:

- Panel d, “expression levels” is too general. Assuming this is \log_2 (normalised counts)? If so, please update. Also applies to Figure 4.

Reply: We thank for the reviewer’s kind suggestions. We updated the y-axis of **figure 3d** and added the statements of “expression levels” in the figure legends of **figure 3** and **figure 4**.

(Page 34): “**Fig. 3d**, The expression levels of each genes in each cell were the normalized, log-transformed ($\log(1+x)$), and scaled (zero mean and unit variance) data.”

(Page 37): “**Fig. 4f-l**, The expression levels of each genes in each cell were the normalized, log-transformed ($\log(1+x)$), and scaled (zero mean and unit variance) data.”

Figure 3d, Violin plot showed the expression levels of outer membrane protein encoding genes (*ompF*, *tsx*, and *lamB*) in samples.

Figure 4:

- Regarding panels f-m, why would a comparison between genes impossible? The scaling can be adjusted to include negative values, and using a common scale across plots like you are showing is common practice. All these plots should use the same scale.

Reply: Thanks for the reviewer’s comments. **Figure 4f-m** were plotted to show the different expression levels among single cells. The expression levels used in panels f-m were the normalized, log-transformed ($\log(1+x)$), and scaled data. The expression level of each gene was scaled to unit variance and zero mean.

We tried to harmonize the labels of **Fig. 4f-m** to -0.5~5 showing as below. However, the UMAP scatter plots with harmonized the labels were not suitable for the visualizations of the different expression levels among single cells or subclusters. For example, the different expression levels of *gldK* and *aceA* among subclusters were indistinguishable in the UMAP scatter plots with the scale of -0.5~5. Thus, we did not make the change of the labels of **Fig. 4f-m**.

Figure 4f-m, Expression levels of marker genes.

Methods:

- For the identification of cell barcodes, authors write they used “reasonable parameters”. This is very unspecific. Why not provide the actual parameters?

Reply: We thank for the reviewer’s comments. We revised the method of the identification of cell barcodes and added the parameters in the manuscript (**Page 20**):

“*Identification of the putative cells*

To determine the number of putative bacteria in each sample, first the total number of detected genes was determined for each possible barcode. Barcodes are ordered from the largest to smallest total number of detected genes and numbered in order. We plotted the $\log_{10}(\text{total number of detected genes per barcode})$ versus $\log_{10}(\text{Barcode number})$ rank plot for each possible barcode. On the rank plot, the knee point (red dashed line) indicated the threshold for putative cells. The knee point was identified according to the calculation results by STAR solo module in STAR (2.7.10a)³, which used the EmptyDrop-like type of filtering, and the 10 numeric parameters: nExpectedCells (1500), maxPercentile (0.99), maxMinRatio (10), indMin (45000), indMax (90000), umiMin (300), umiMinFracMedian (0.01), candMaxN (20000), FDR (0.01), simN (10000). Barcodes on the left of the threshold were identified as bacteria, but the rest was identified as noise. Only the barcodes identified as bacteria were used for downstream analysis.”

- Saturation analysis – how were they randomly sub-sampled? Also, the saturation section should be moved to after “Analysis of gene expression matrix”.

Reply: Thanks for the reviewer’s comments. We revised the method of saturation analysis and added the code for how reads were randomly sub-sampling. We moved the saturation analysis section to after “Analysis of gene expression matrix” in the revised manuscript (**Page 22**):

“*Saturation analysis*

The rRNA depleted and deeper sequenced *E. coli* sample were used to generate the saturation curve. Saturation curve plot was generated by randomly selecting the corresponding number of raw reads from the sample library and then using the same alignment pipeline to calculate the

median numbers of genes detected per cell. Sequencing reads were randomly subsampled to 22k, 18k, 13k, 9k, 4k, 2k, 1k reads per cell using the Seqtk tool.”

Readme file: “Fig.2h: Saturation analysis.sh”

Supplementary Figure 7:

- Thanks for including these two plots. The right-hand side plot is very informative and highlights the 5’-3’ read coverage of your method. Panel a, however, is not that informative due to the sparse nature of sequencing single cells, so it could be removed. Naturally, this decision is up to the authors.

Reply: Thanks for the reviewer’s comments. We removed the panel a in supplementary figure 7.

Reviewer #3 (Remarks to the Author):

Overall, the authors have done a good job at addressing reviewer comments but I still have a major concern.

Reply: We are grateful to reviewer's effort to review our paper. We have fully addressed the concerns as follows.

The method of this paper is clearly very sensitive and seems to result in very good UMI counts across many species. However, in some instances, the numbers reported are really extraordinarily high to the point that I find them difficult to believe without some additional analysis or discussion.

*For *B. subtilis* the authors report detecting 6564 umis from 1249 genes in 45 cells. The UMI numbers are about 10-20x higher than any previously reported. My concern is that these extremely high numbers are some sort of artifact resulting from cell clumps, very unusual growth conditions, PCR or data analysis.*

Reply: We thank for the reviewer's comments. We used vortex oscillator to make single cell suspension during preprocessing. We confirmed the single bacterial morphology under a microscope (**Supplementary Fig. 1a**) before proceeding to microfluidic encapsulation. The *B. subtilis* were cultured using routine method and sampled upon reaching the exponential phase (OD₆₀₀ ~0.5). We also checked the PCR and data analysis and did not find anything wrong.

We have discussed the high efficiency of smRandom-seq in the discussion section. smRandom-seq was developed with several optimized strategies, including the random primer design for efficient reverse transcription, the dA tailing strategy for efficient second strand cDNA synthesis, the droplet microfluidics platform for single bacterium barcoding. We also optimized the droplet barcoding platform by generating smaller barcoded beads and droplets, and modifying the primers releasing and the barcodes synthesis strategy.

Please perform at least the following quality controls:

*1. Supplementary Fig. 3 shows that the mRNA UMI counts in *E. coli* are ~3x lower than in *B. subtilis* but that rRNA counts are 3x higher. Is it possible that some *B. subtilis* rRNA genes are incorrectly labeled as mRNA? Please carefully check gene annotations and functions.*

Reply: Thanks for the reviewer's careful reading. We checked the gene annotations and functions for the *E. coli* and *B. subtilis* mixture and did not found anything wrong. We checked the other published datasets and found that microSPLiT also reported lower mRNA counts but higher rRNA counts in *E. coli* than *B. subtilis*⁴.

The *E. coli* and *B. subtilis* species-mixing experiment by MicroSPLIT

(C) mRNA and rRNA UMI counts per cell for both species.

2. Include a gene expression analysis to show which specific genes are most highly expressed and how many mRNA are detected for these genes. Please also consider correlating your gene expression data to previous bulk or single cell datasets for *B. subtilis*. While this analysis cannot address issues arising from cell clumping, it would alleviate concerns relating to the analysis pipeline or to unusual growth conditions.

Reply: Thanks for the reviewer's valuable comments. We correlated our single bacterium RNA-seq data to the public bulk RNA-seq dataset⁵ for exponential *B. subtilis* (GSM6729945 Emp3h i Exponential growth in GSE217916) and calculated the top 10 highest expression genes, respectively (**Supplementary fig. 3f**). There were 5 overlapped gene IDs between the top 10 highest expression genes of our single bacterium and the previous bulk RNA-seq dataset.

The top 10 highest expression genes in B. subtilis .			
smRandom-seq		Public bulk RNA-seq	
Gene ID	Count proportion	Gene ID	Count proportion
B4U62_00715	0.032357487	B4U62_00710	0.021309632
B4U62_00710	0.024757256	B4U62_00715	0.020266442
B4U62_21090	0.017682124	B4U62_00870	0.009370901
B4U62_00745	0.015061197	B4U62_03460	0.009284874
B4U62_16755	0.012081833	B4U62_00830	0.009189792
B4U62_00870	0.00932849	B4U62_21090	0.008860777
B4U62_00485	0.009104753	B4U62_02035	0.008415549
B4U62_03370	0.00902028	B4U62_00745	0.008308393
B4U62_00700	0.00899745	B4U62_10560	0.007835396
B4U62_19050	0.008785127	B4U62_00690	0.007555582

Supplementary fig. 3f, The top 10 highest expression genes sorted by the count proportion in *B. subtilis* datasets of the species mixture by smRandom-seq and the public bulk RNA-seq dataset.

3. Another observation is that the UMI and gene counts are generated from a very small number of 45 cells. What criterion was used to pick such a small number? Is there a graph similar to 2g for *B. subtilis*?

Reply: Thanks for the reviewer's comments. We processed and collected a small number of cells for sequencing in the species-mixture experiment, so that a deep sequencing depth could be

achieved with low sequencing cost at the initial stage of our study. We added the UMI count vs. barcode plot (**Supplementary 3a**) for cell number identification of the *E. coli* and *B. subtilis* mixture.

Supplementary 3a, UMI count versus barcode rank plot of the *E. coli* and *B. subtilis* mixture.

Another example that I find surprising is the data shown in Fig. S11 where *E. coli* UMIs > 2000 are shown which is more than even in the species mixing experiment. Again, it would be helpful to indicate which genes drive this expression. What are the top 10 expressed genes at $t=0$ and how do they compare to the other *E. coli* datasets in the same paper and prior work?

Reply: We thank for the reviewer's comment. We compared the *E. coli* dataset in **Fig. S11** (CIP-T0 by smRandom-seq) to the other *E. coli* datasets in the same paper (species-mixture by smRandom-seq in **figure 2a**) and prior work (public bulk RNA-seq dataset: GSM5172890 BW25113 Wild Type ECWT1_1 in GSE168963) and calculated the top 10 highest expression genes, respectively (**Supplementary fig. 11c**). There were 5 overlapped gene IDs among the top 10 highest expression genes of the *E. coli* datasets in **Fig. S11** (the CIP-T0 sample), the same paper (the species-mixture in **figure 2a**) and prior work (the public bulk RNA-seq dataset⁶).

The top 10 highest expression genes in *E. coli*.

CIP-T0 by smRandom-seq		Species-mixture by smRandom-seq		Public bulk RNA-seq	
Gene ID	Count proportion	Gene ID	Count proportion	Gene ID	Count proportion
BW25113_RS19530	0.1785308	BW25113_RS19995	0.145887	BW25113_RS19530	0.30111
BW25113_RS01010	0.149809035	BW25113_RS19530	0.066561	BW25113_RS19995	0.211552
BW25113_RS23140	0.101970708	BW25113_RS01025	0.060744	BW25113_RS01025	0.211342
BW25113_RS13535	0.073189522	BW25113_RS19520	0.043218	BW25113_RS13535	0.064297
BW25113_RS01025	0.065432498	BW25113_RS13535	0.042552	BW25113_RS19520	0.058034
BW25113_RS23125	0.055954742	BW25113_RS04685	0.012936	BW25113_RS19980	0.039734
BW25113_RS19995	0.041493869	BW25113_RS17310	0.012513	BW25113_RS17000	0.038523
BW25113_RS13525	0.036868505	BW25113_RS01010	0.012182	BW25113_RS16985	0.029103
BW25113_RS22750	0.030024506	BW25113_RS09310	0.011086	BW25113_RS01010	0.024043
BW25113_RS16985	0.027230109	BW25113_RS00570	0.009709	BW25113_RS23125	0.00807

Supplementary fig. 11c, The top 10 highest expression genes sorted by the count proportion in *E. coli* datasets of the CIP-T0 sample, the species mixture by smRandom-seq and the public bulk RNA-seq dataset.

1. McNulty, R. et al. Probe-based bacterial single-cell RNA sequencing predicts toxin regulation. *Nature Microbiology* **8**, 934-945 (2023).
2. Homberger, C., Hayward, R.J., Barquist, L. & Vogel, J. Improved Bacterial Single-Cell RNA-Seq through Automated MATQ-Seq and Cas9-Based Removal of rRNA Reads. *mBio* **14**, e03557-03522 (2023).
3. Dobin, A. et al. STAR: ultrafast universal RNA-seq aligner. *Bioinformatics* **29**, 15-21 (2013).
4. Kuchina, A. et al. Microbial single-cell RNA sequencing by split-pool barcoding. *Science* **371** (2021).
5. Wang B, K.F., Hamoen LW Induction of the CtsR regulon improves Xylanase production in *Bacillus subtilis*. *Preprint from Research Square* (2023).
6. Ojo, O., Scott, D., Iwalokun, B., Odetoyin, B. & Grove, A. Transcriptome RNA Sequencing Data Set of Differential Gene Expression in *Escherichia coli* BW25113 Wild-Type and slyA Mutant Strains. *Microbiol Resour Announc* **10** (2021).

Reviewers' Comments:

Reviewer #3:

Remarks to the Author:

All my concerns were addressed in this revision and the paper can be published as is.